# Bilevel Coreset Selection in Continual Learning: A New Formulation and Algorithm

**Jie Hao**
Department of Computer Science
George Mason University
`jhao6@gmu.edu`

**Kaiyi Ji**
Department of CSE
University at Buffalo
`kaiyiji@buffalo.edu`

**Mingrui Liu**[*]
Department of Computer Science
George Mason University
`mingruil@gmu.edu`

## Abstract

Coreset is a small set that provides a data summary for a large dataset, such that training solely on the small set achieves competitive performance compared with a large dataset. In rehearsal-based continual learning, the coreset is typically used in the memory replay buffer to stand for representative samples in previous tasks, and the coreset selection procedure is typically formulated as a bilevel problem. However, the typical bilevel formulation for coreset selection explicitly performs optimization over discrete decision variables with greedy search, which is computationally expensive. Several works consider other formulations to address this issue, but they ignore the nested nature of bilevel optimization problems and may not solve the bilevel coreset selection problem accurately. To address these issues, we propose a new bilevel formulation, where the inner problem tries to find a model which minimizes the expected training error sampled from a given probability distribution, and the outer problem aims to learn the probability distribution with approximately $K$ (coreset size) nonzero entries such that learned model in the inner problem minimizes the training error over the whole data. To ensure the learned probability has approximately $K$ nonzero entries, we introduce a novel regularizer based on the smoothed top-$K$ loss in the upper problem. We design a new optimization algorithm that provably converges to the $\epsilon$-stationary point with $O(1/\epsilon^4)$ computational complexity. We conduct extensive experiments in various settings in continual learning, including balanced data, imbalanced data, and label noise, to show that our proposed formulation and new algorithm significantly outperform competitive baselines. From bilevel optimization point of view, our algorithm significantly improves the vanilla greedy coreset selection method in terms of running time on continual learning benchmark datasets. The code is available at `https://github.com/MingruiLiu-ML-Lab/Bilevel-Coreset-Selection-via-Regularization`.

## 1   Introduction

Deep Neural Networks (DNNs) have achieved tremendous successes in various domains, including computer vision [41, 30], natural language processing [72, 15], generative modeling [26] and

---

[*]Corresponding Author.

37th Conference on Neural Information Processing Systems (NeurIPS 2023).

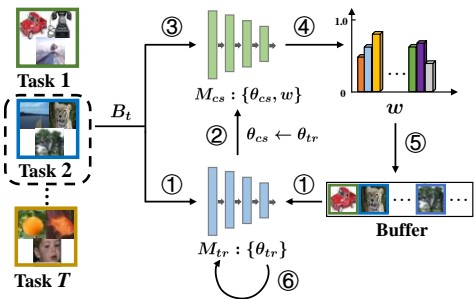

Figure 1: Illustration of our algorithm. There are two neural network models, one is $M_{tr}$ for model training, and the other is $M_{cs}$ for coreset selection. A coreset is selected from the current data stream $B_t$ by conducting six steps. ① Feed a stream mini-batch $B_t$ and sampled buffer data to $M_{tr}$. ② Copy model parameters from $M_{tr}$ to $M_{cs}$: $\theta_{cs} \leftarrow \theta_{tr}$. ③ Feed a mini-batch $B_t$ into model $M_{cs}$. ④ Conduct bilevel optimization to update the model parameter $\theta_{cs}$ in $M_{cs}$ and output a probability distribution $w$. ⑤ Sample a coreset from $B_t$ based on the distribution of $w$ and add the sampled data into buffer. ⑥ Calculate stochastic gradient based on $B_t$ and sampled data in the buffer in Step 1, and update $\theta_{tr}$ based on gradient information. Repeat the above steps for each stream mini-batch.

games [68]. However, in continual learning, where DNNs are trained on a sequence of tasks with possibly non-i.i.d. data, the performance will be degraded on the previously trained tasks. This is referred to as *catastrophic forgetting* [53, 52, 60]. To alleviate catastrophic forgetting, one of the effective ways is *rehearsal-based continual learning*, where a small replay buffer is maintained and revisited during the continuous learning process. There is a line of works studying how to efficiently maintain the replay buffer using the coreset selection approach [6, 78, 83], in which a small set of data is selected as representative samples to be used in continual learning.

The coreset selection in continual learning is formulated as a cardinality-constrained bilevel optimization problem which is solved by incremental subset selection [6]. This greedy approach is computationally expensive and hence is not scalable when the coreset size is large. To address this issue, Zhou et al. [83] propose a relaxation of the bilevel formulation in [6], which drops the nested nature of bilevel formulation and actually becomes two sequential optimization problems. Tiwari et al.proposed a gradient approximation method in [71], which selects a coreset that approximates the gradient of model parameters over the entirety of the data seen so far. Yoon et al. [78] proposes an online coreset selection method by maximizing several similarity metrics based on data pairs within each minibatch, and sample pairs between each minibatch and coreset. These approaches do not directly address the algorithmic challenges caused by the nested nature of the bilevel optimization problem, and may not solve the original bilevel coreset selection problem efficiently.

The key challenges in the bilevel coreset selection problems are two folds. First, the bilevel formulation in [6] needs to directly perform optimization over cardinality constraint, which is a nonconvex set and greedy approaches are expensive when the coreset size is large. Second, the bilevel formulation in [6] has a nested structure: one problem is embedded within another, and the outer and inner functions both have dependencies over the same set of decision variables. It remains unclear how to design efficient algorithms to solve constrained bilevel optimization algorithms for the coreset selection problem with provable theoretical guarantees.

Our proposed solution addresses these challenges with a novel bilevel formulation and provably efficient optimization algorithms. The proposed new bilevel formulation is referred to as *Bilevel Coreset Selection via Regularization* (BCSR). The main differences between our approach and the standard bilevel approach in [6] are: (i) unlike the standard bilevel formulation which requires performing optimization based on a cardinality constraint, we propose to solve a bilevel optimization on a probability simplex over training examples; (ii) to make sure the probability distribution lies in a low dimensional manifold, we propose to add a smoothed top-$K$ loss as a regularizer to the upper-level problem; (iii) due to our new formulation, we are able to design a simple and effective first-order method to solve this new bilevel problem with provable non-asymptotic convergence guarantees. The first-order method is easy to implement and much faster than the greedy approach as in [6]. Our main contribution is listed as follows.

**Algorithm 1** PyTorch-style pseudocode for BCSR

```
1  # tasks: the task sequences; T: the number of tasks
2  # L(): loss function; x: image batch, y: label batch
3  # model, model_proxy: for training and coreset selection
4  # update_w: a function updating outer variable w
5  M = [[] * T]    # memory buffer
6  for ind, task in tasks:
7      for (x, y) in dataloader:
8          model_proxy.load_state_dict(model.parameters())
9          (x_m, y_m) = next_batch(M)
10         out1 = model(x)
11         out2 = model(x_m)
12         loss = L (out1, y)+ L (out2, y_m)
13         loss.backward()
14         (x_core, y_core) = find_coreset(x, y, K, model_proxy)
15         M[ind].append((x_core, y_core))
16         update(model.params)
17 def find_coreset(x, y, K, model_proxy):
18     coreset_w = 1.0/y.size()
19     for j in range(outer_loops):
20         for i in range(inner_loops):
21             lower_loss =  L (coreset_w * model (x, y) )
22             lower_loss.backward()
23             update(model_proxy.params)
24         coreset_w = update_w(coreset_w)
25     return torch.multinomial(coreset_w , K)
```

- We propose a new bilevel formulation, namely BCSR, for the coreset selection in rehearsal-based continual learning. Instead of directly learning the binary masks for each sample, the new formulation tries to learn a probability distribution in a low-dimensional manifold by adding a smoothed top-$K$ loss as a regularizer in the upper problem. This formulation is designed to satisfy two important features in continual learning with DNNs: (i) keeping the nested structure in the coreset selection; (ii) being amenable to first-order algorithms, which makes it easy to implement in modern deep learning frameworks such as PyTorch and TensorFlow. Based on the new formulation, we propose an efficient first-order algorithm for solving it. The main workflow of our algorithm is illustrated in Figure 1, and the corresponding Pytorch-style pseudocode is presented in Algorithm 1.

- We have conducted extensive experiments among various scenarios to verify the effectiveness of our proposed algorithm, including balanced, imbalanced, and label-noise data. Our algorithm outperforms all baselines for all settings in terms of average accuracy, and it is much better than all other coreset selection algorithms. For example, on imbalanced data of Multiple Datasets, BCSR is better than the best coreset selection algorithm by $4.65\%$ in average accuracy. From bilevel optimization point of view, our algorithm significantly improves the vanilla greedy coreset selection method [6] in terms of running time on continual learning benchmark datasets.

- Under the standard smoothness assumptions of the loss function, we show that our algorithm requires at most $O(1/\epsilon^4)$ complexity for finding an $\epsilon$-stationary point in the constrained case[2]. Notably, the $O(1/\epsilon^4)$ complexity consists of $O(1/\epsilon^2)$ backpropagations and $O(1/\epsilon^4)$ samplings from Gaussian distribution, where the latter cost is computationally cheap.

## 2  Related Work

**Continual Learning**  There are different classes of continual learning methods, including regularization-based approaches [39, 81, 10, 1, 59, 64, 17], dynamic architecture methods [61,

---

[2]In the constrained setting, the definition of $\epsilon$-stationary point is defined with gradient mapping, i.e., $w$ is a $\epsilon$-stationary point of the function $\phi$ if $\frac{1}{\beta}\|w - \mathcal{P}_\Delta(w - \beta\nabla\phi(w))\| \le \epsilon$, where $\beta$ is the stepsize, $\mathcal{P}$ is the projection operator, $\Delta$ is the probability simplex. This matches the best iteration complexity as in the single level optimization problem [24].

79, 62, 51, 76, 44, 77], and rehearsal-based methods [48, 57, 11, 58, 31, 3, 18, 28, 80, 6, 83, 78, 84]. In the rehearsal-based continual learning, the memory is either reproduced experience replay [48] or generative replay [67]. Our work focuses on the aspect of the coreset selection on the replay memory and can be flexibly integrated into rehearsal-based methods in continual learning.

**Coreset Selection**   The coreset selection methods were used frequently in supervised and unsupervised learning, such as $k$-means [19], Gaussian mixture model [49], logistic regression [33] and bayesian inference [9]. They were also used frequently in the active learning literature [74, 63]. The coreset selection in continual learning is related to the sample selection [34, 2, 3]. Nguyen et al. [55] introduce variational continual learning which is combined with coreset summarization [5]. Borsos et al. [6] proposed the first bilevel formulation for the coreset selection in continual learning, which is later improved by [83, 78]. Compared with these works, our work focuses on improved bilevel coreset selection: we provide a better bilevel formulation than [6] and design a provably efficient optimization algorithm.

**Bilevel optimization**   Bilevel optimization is used to model nested structure in the decision-making process [73]. Recently, gradient-based bilevel optimization methods have broad applications in machine learning, including meta-learning [20], hyperparameter optimization [56, 22], neural architecture search [46], and reinforcement learning [40, 32]. These methods can be generally categorized into implicit differentiation [16, 56, 45, 4] and iterative differentiation [50, 21, 20, 65, 27] based approaches. Recently, various stochastic bilevel algorithms have been also proposed and analyzed by [12, 37, 25, 32, 29, 4, 14]. A comprehensive introduction can be found in the survey [47]. In this work, we propose a novel stochastic bilevel optimizer with very flexible parameter selection, which shows great promise in the coreset selection for continual learning.

## 3   New Bilevel Formulation for Coreset Selection in Continual Learning

In this section, we first introduce our new bilevel formulation, namely Bilevel Coreset Selection via Regularization (BCSR). The key idea of this approach is to learn a probability distribution over the whole dataset such that the best model parameter obtained by minimizing the loss on the sampled dataset (i.e., the minimizer for the lower-level problem) is also the best for the whole dataset (i.e., the minimizer for the upper-level problem), and then a coreset can be sampled based on the learned probability distribution. In addition, the learned probability distribution is expected to lie in a low dimensional manifold (i.e., with $K$ nonzero entries where $K$ is the coreset size). To achieve this, we added a smooth top-$K$ loss as a regularizer to promote the probability distribution to have $K$ nonzero entries. Specifically, the objective function of BCSR is:

$$\min_{\substack{0 \leq w_{(i)} \leq 1 \\ ||w||_1 = 1}} \left[ \phi(w) = \sum_{i=1}^{n} \ell_i(\theta^*(w)) - \lambda \sum_{i=1}^{K} \mathbb{E}_z (w + \delta z)_{[i]} \right]$$

$$s.t., \theta^*(w) = \arg\min_{\theta} \left[ L(\theta, w) = \sum_{i=1}^{n} w_{(i)} \ell_i(\theta) \right] \tag{1}$$

where $n$ is the sample size, $\theta$ is the model parameter, $w$ is the sample weights, $\ell_i(\theta)$ denote the loss function calculated based on $i$-th sample with model parameter $\theta$, $w_{(i)}$ is the $i$-th coordinate of $w$, $w_{[i]}$ is the $i$-th largest component of $w$, $\lambda > 0$ is the regularization parameter, $w + \delta z$ denote to adding $\delta z$ on each coordinate of $w$ where $z \sim \mathcal{N}(0, 1)$. Note that $R(w, \delta) := -\lambda \sum_{i=1}^{K} \mathbb{E}_z (w + \delta z)_{[i]}$ denote the smoothed top-$K$ regularization. We add this regularization to make sure the summation of the top-$K$ entries of the learned probability vector is large, such that we can confidently choose a coreset with size $K$. The goal of employing Gaussian noise to the regularizer is for the ease of algorithm design: this Gaussian smoothing technique can make the regularizer to be smooth such that it is easier to design efficient first-order bilevel optimization solvers. Otherwise, the upper-level problem would become nonconvex and nonsmooth, and it would be difficult for algorithm design under this case.

**Discussion and Comparison with Prior Works**   In this part, we illustrate how this new formulation addresses the drawbacks of the previous approaches. The work of [6] does not use any regularizer and regards the weight of each sample as a binary mask. This formulation needs to solve a combinatorial

---

**Algorithm 2** Bilevel Coreset Selection via Regularization (BCSR)

---

**Input**: Dataset $\mathcal{D}$
**Initialize**: model parameter $\theta_0$, memory $\mathcal{M} = \{\}$

  1: **for** batch $\mathcal{B}_t \sim \mathcal{D}$ **do**
  2:     Compute coreset $\mathcal{S}_t = \text{Find-coreset}(\theta_{t-1}, \mathcal{B}_t)$
  3:     $\mathcal{M} = \mathcal{M} \cup \mathcal{S}_t$
  4: **end for**

---

---

**Algorithm 3** Find-coreset$(\theta, \mathcal{B})$

---

**Input**: the current model parameter $\theta$, the current batch $\mathcal{B}$, the iteration parameters $J, N, Q$
**Initialize**: coreset size $K$, $n = |\mathcal{B}|$, and $v_0$

  1: $w_0 = [\frac{1}{n}, \ldots, \frac{1}{n}]$ (uniform probability initialization)
  2: $\theta_1^0 = \theta$
  3: **for** $j = 0, 2, \ldots, J - 1$ **do**
  4:     Set initialization $\theta_j^0 = \theta_{j-1}^N$ if $j > 0$ and $\theta_1^0$ otherwise
  5:     **for** $k = 1, \ldots, N$ **do**
  6:       update $\theta_j^k$ according to eq. (2).
  7:     **end for**
  8:     Set initialization $v_j^0 = v_{j-1}^Q$ if $j > 0$ and $v_0$ otherwise
  9:     Compute estimate $v_j^Q$ by solving QP in (4) by GD with stepsize $\eta$ and inital $v_j^0 = v_{j-1}^Q$.
10:     Compute hypergradient estimate in (3)
11:     Update $w_{j+1}$ and project it onto simplex by (5)
12: **end for**
13: $\mathcal{S} \leftarrow$ a set of $K$ data points sampled from $\mathcal{B}$ according to the probability distribution $w_J$
14: Return $\mathcal{S}$

---

optimization problem and their approach of incremental subset selection is computationally expensive. The work of [83] relaxes the bilevel formulation in [6] to minimize the expected loss function over the Bernoulli distribution $s$, i.e., $\min_{s \in \mathcal{C}} \Phi(s)$, and develops a policy gradient solver to optimize the Bernoulli variable. Their gradient $\nabla_s \Phi(s) = \mathbb{E}_{p(m|s)} L(\theta^*(m)) \nabla_s \ln p(m|s)$ does not include the implicit gradient of $L(\theta^*(m))$ in terms of $s$. However, $\theta^*(m)$ actually depends on the mask $m$, and $m$ depends on the Bernoulli variable $s$. In contrast, our bilevel optimization computes the hypergradients for the coreset weights $w$ ($0 \leq w \leq 1$ and $\|w\|_1 = 1$), which considers the dependence between $\theta(w)$ and $w$ [3]. In addition, Zhou et al. [83] assume that the inner loop can obtain the exact minimizer $\theta^*(m)$, which may not hold in practice. In contrast, we carefully analyze the gap between the estimated $\theta^*(w)$ and itself by our algorithm and analysis.

## 4 Algorithm Design

Equipped with the new formulation, the entire algorithm is presented in Algorithm 2, which calls Algorithm 3 as a subroutine. Each time the algorithm encounters a minibatch $\mathcal{B}$, a coreset is selected within this minibatch by invoking Algorithm 3. Algorithm 3 is a first-order algorithm for solving the bilevel formulation (1). In Algorithm 3, the model parameter $\theta$ and the weight distribution $w$ are updated alternatively. We first perform $N$ steps of gradient descent steps to find a sufficiently good $\theta$ for the lower-level problem (lines 5-7) by the update rule:

$$\theta_j^k = \theta_j^{k-1} - \alpha \nabla_\theta L(\theta_j^{k-1}, w_j), \tag{2}$$

where $\theta_j^k$ denotes the model parameters at the $j$-th outer loop and the $k$-th inner loop. To update the outer variable $w$, BCSR approximates the true gradient $\nabla \phi(w)$ of the outer function w.r.t $w$, which is called hypergradient [56]. BCSR constructs a hypergradient estimator:

$$\varphi_j = \frac{1}{|\mathcal{B}|} \sum_{\widetilde{z} \in \mathcal{B}} \nabla_w R(w, \delta; \widetilde{z}) - \nabla_w \nabla_\theta L(\theta_j^N, w_j) \left[ (\nabla_\theta^2 L(\theta_j^N, w_j))^{-1} (\sum_{i=1}^n \nabla_\theta \ell_i(\theta_j^N)) \right], \tag{3}$$

---

[3]The coreset weight $w$ in our formulation is equivalent to sample mask $s$ in [83]

Table 1: Experiment results on Split CIFAR-100

| Methods | Balanced | | Imbalanced | | Label Noise | |
|---|---|---|---|---|---|---|
| | $A_T$ | $FGT_T$ | $A_T$ | $FGT_T$ | $A_T$ | $FGT_T$ |
| K-means Features | 57.82±0.69 | 0.070±0.003 | 45.44±0.76 | 0.037±0.002 | 57.38±1.26 | 0.098±0.003 |
| K-means Embedding | 59.77±0.24 | 0.061±0.001 | 43.91±0.15 | 0.044±0.001 | 57.92±1.25 | 0.091±0.016 |
| Uniform | 58.99±0.54 | 0.074±0.004 | 44.73±0.11 | 0.033±0.007 | 58.76±1.07 | 0.087±0.006 |
| iCaRL | 60.74±0.09 | **0.044±0.026** | 44.25±2.04 | 0.042±0.019 | 59.70±0.70 | 0.071±0.010 |
| Grad Matching | 59.17±0.38 | 0.067±0.003 | 45.44±0.64 | 0.038±0.001 | 59.58±0.28 | 0.073±0.008 |
| SPR | 59.56±0.73 | 0.143±0.064 | 44.45±0.55 | 0.086±0.023 | 58.74±0.63 | 0.073±0.010 |
| MetaSP | 60.14±0.25 | 0.056±0.230 | 43.74±0.36 | 0.079±0.014 | 57.43±0.54 | 0.086±0.007 |
| Greedy Coreset | 59.39±0.16 | 0.066±0.017 | 43.80±0.01 | 0.039±0.007 | 58.22±0.16 | 0.066±0.001 |
| GCR | 58.73±0.43 | 0.073±0.013 | 44.48±0.05 | 0.035±0.005 | 58.72±0.63 | 0.081±0.005 |
| PBCS | 55.64±2.26 | 0.062±0.001 | 39.87±1.12 | 0.076±0.011 | 56.93±0.14 | 0.100±0.003 |
| OCS | 52.57±0.37 | 0.088±0.001 | 46.54±0.34 | 0.022±0.003 | 51.77±0.81 | 0.103±0.007 |
| BCSR | **61.60±0.14** | 0.051±0.015 | **47.30±0.57** | **0.022±0.005** | **60.70±0.08** | **0.059±0.013** |

where $R(w, \delta; \widetilde{z}) := -\lambda \sum_{i=1}^{K} (w + \delta \widetilde{z})_{[i]}$ and $\widetilde{z} \sim \mathcal{N}(0, 1)$. Solving the Hessian-inverse-vector product in eq. (3) is computationally intractable. We denote $v^* := (\nabla_\theta^2 L(\theta_j^N, w_j))^{-1} (\sum_{i=1}^{n} \nabla_\theta \ell_i(\theta_j^N))$ in eq. (3), where $v^*$ can be approximated by solving the following quadratic programming problem efficiently by $Q$ steps of gradient descent (line 9):

$$\min_v \frac{1}{2} v^T \nabla_\theta^2 L(\theta_j^N, w_j) v - v^T \sum_{i=1}^{n} \nabla_\theta \ell_i(\theta_j^N). \tag{4}$$

Next, the hypergradient estimate (line 10) is computed based on the output of approximated quadratic programming. Note both model parameters and sample weights need to use warm start initialization (line 4 and line 8). Then the weight is updated and projected onto simplex (line 11):

$$\hat{w}_{j+1} = w_j - \beta \varphi_j, \ \ w_{j+1} = \mathcal{P}_{\Delta^n}(\hat{w}_{j+1}), \tag{5}$$

where $\Delta^n := \{w \in \mathbb{R}^n : 0 \le w_{(i)} \le 1, \|w\|_1 = 1\}$. In terms of other experimental hyperparameters, we allow very flexible choices of hyperparameters (e.g., $N$, $Q$) as shown in our theory to achieve polynomial time complexity for finding a $\epsilon$-stationary point.

The selected coresets for each task are stored in a memory buffer with a fixed size $m$. There is a separated memory slot for each task with size $[m/i]$ when the task $i$ comes. After each task $i$, the memory slots before $i$ will randomly remove some samples to adjust all the memory slots to $[m/i]$. That means the memory size for each task decreases as the task ID increase to maintain the total buffer size $m$ unchanged. The same memory strategy is also used in the greedy coreset approach [6].

## 5 Experiments

We conduct extensive experiments under various settings, including balanced data, imbalanced data, and label-noise data. The empirical results demonstrate the effectiveness of our method in rehearsal-based continual learning.

### 5.1 Experimental Setup

**Datasets** We use commonly-used datasets in the field of continual learning, including Split CIFAR-100, Permuted MNIST, Multiple Datasets, Tiny-ImageNet, and Split Food-101. We follow the experimental settings as that in prior work [78] and [83]. Each dataset is processed with three approaches: balanced, imbalanced, and label-noise. Please refer to Appendix L for more details about data processing and settings.

**Baselines** We compare our algorithm BCSR with other continual learning methods based on coreset strategy, including $k$-means features [55], $k$-means embedding [63], Uniform Sampling, iCaRL [57], Grad Matching [9], Greedy Coreset [6], PBCS [83], GCR [71], and OCS [78]. We also compare with non-coreset reply method, SPR [38], MetaSP [70]. All algorithms are built upon episodic memory, which stores coreset selected from stream data. Then a model, such as ResNet-18 [30], is trained over the data from the current stream and the episodic memory.

**Metrics** Average accuracy and forgetting measure [10] are two primary evaluation metrics that are used in continual learning literature. AVG ACC ($A_T$) is the average accuracy tested on all tasks after finishing the task $T$: $A_T = \frac{1}{T} \sum_{i=1}^{T} a_{T,i}$, where $a_{T,i}$ is the test accuracy of task $i$ after training

Table 2: Experiment results on Multiple Datasets

| Methods | Balanced | | Imbalanced | | Label Noise | |
|---|---|---|---|---|---|---|
| | $A_T$ | $FGT_T$ | $A_T$ | $FGT_T$ | $A_T$ | $FGT_T$ |
| K-means Features | 54.63±0.88 | 0.138±0.007 | 33.63±2.66 | 0.136±0.063 | 45.46±3.50 | 0.120±0.049 |
| K-means Embedding | 56.83±1.65 | 0.106±0.019 | 35.93±1.60 | 0.106±0.031 | 46.32±3.19 | 0.084±0.030 |
| Uniform | 55.93±0.03 | 0.101±0.032 | 35.48±2.96 | 0.104±0.025 | 48.68±0.44 | 0.079±0.002 |
| iCaRL | 56.19±0.32 | 0.130±0.012 | 42.18±1.59 | 0.057±0.022 | 49.22±0.54 | 0.067±0.010 |
| Grad Matching | 53.41±0.46 | 0.119±0.020 | 38.16±3.90 | 0.082±0.003 | 46.96±1.05 | 0.091±0.029 |
| SPR | 56.20±1.91 | 0.124±0.036 | 40.79±1.73 | 0.143±0.051 | 49.77±1.58 | 0.062±0.024 |
| MetaSP | 57.14±1.10 | 0.113±0.042 | 41.32±1.50 | 0.103±0.053 | 47.14±1.66 | 0.081±0.027 |
| Greedy Coreset | 53.56±0.06 | 0.099±0.005 | 22.57±1.10 | 0.265±0.022 | 41.32±1.51 | 0.137±0.009 |
| GCR | 54.35±0.31 | 0.125±0.014 | 35.13±2.79 | 0.105±0.043 | 47.58±1.30 | 0.078±0.016 |
| PBCS | 52.93±0.28 | 0.152±0.016 | 37.25±2.93 | 0.115±0.033 | 47.51±1.56 | 0.101±0.021 |
| OCS | 55.65±2.26 | **0.062±0.001** | 40.48±1.39 | 0.051±0.003 | 45.03±4.16 | **0.049±0.012** |
| BCSR | **59.89±0.95** | 0.096±0.005 | **45.13±0.54** | **0.046±0.008** | **49.97±1.14** | 0.064±0.031 |

Table 3: Experiment results on Tiny-ImageNet

| Methods | Balanced | | Imbalanced | | Label Noise | |
|---|---|---|---|---|---|---|
| | AVG ACC | FGT | AVG ACC | FGT | AVG ACC | FGT |
| K-means Features | 41.20±0.75 | 0.131±0.004 | 36.27±0.30 | 0.079±0.014 | 36.68±1.35 | 0.095±0.004 |
| K-means Embedding | 41.48±1.21 | 0.129±0.007 | 36.29±0.23 | 0.085±0.003 | 36.01±1.51 | 0.083±0.005 |
| Uniform | 42.11±0.52 | 0.129±0.002 | 37.07±0.53 | 0.083±0.009 | 37.14±1.05 | 0.099±0.003 |
| iCaRL | 43.84±0.09 | 0.114±0.004 | 37.65±0.84 | 0.058±0.003 | 38.52±0.25 | 0.063±0.006 |
| Grad Matching | 43.45±0.32 | 0.105±0.007 | 37.58±0.39 | 0.066±0.004 | 38.84±0.42 | 0.064±0.006 |
| SPR | 42.79±0.50 | **0.102±0.009** | 36.55±0.74 | 0.070±0.026 | 39.89±0.53 | 0.065±0.021 |
| MetaSP | 43.33±0.32 | 0.127±0.002 | 36.75±0.57 | 0.086±0.006 | 37.18±0.76 | 0.068±0.007 |
| Greedy Coreset | 41.02±0.33 | 0.119±0.017 | 33.43±0.86 | 0.103±0.002 | 36.37±0.16 | 0.079±0.006 |
| GCR | 41.45±0.35 | 0.125±0.008 | 36.08±0.62 | 0.072±0.017 | 37.46±0.40 | 0.115±0.011 |
| PBCS | 36.99±0.15 | 0.177±0.002 | 35.88±0.16 | 0.071±0.080 | 37.02±0.16 | 0.133±0.029 |
| OCS | 41.29±0.09 | 0.112±0.001 | 35.09±0.60 | **0.036±0.011** | 35.36±0.94 | 0.061±0.005 |
| BCSR | **44.13±0.33** | 0.106±0.001 | **38.59±0.11** | 0.070±0.004 | **40.72±0.56** | **0.055±0.006** |

the task $T$. $FGT_T$ evaluates the performance drop on the past tasks after training on the task $T$:
$$FGT_T = \frac{1}{T} \sum_{j=1}^{T} \left[ \max_{j \in 1, \dots T-1} \left( a_{j,i} - a_{T,i} \right) \right].$$

**Implementation Details** Following the implementation of the coreset-based algorithm [6], we use two models for the purposes of model training and coreset selection respectively, where the first model is denoted as $M_{tr}$ for model training, and the second model (also known as a proxy model) is denoted as $M_{cs}$ for coreset selection. In the section of the model training, we adopt a single-head MLP with two hidden layers for Permuted MNIST, and a ResNet-18 for Split CIFAR-100 and other datasets. In the section of the coreset selection, $M_{cs}$ adopts the same architecture as $M_{tr}$. This is the key difference between our work and the Greedy Coreset [6]: they use Neural Tangent Kernels (NTK) [35] while we use a specific deep neural network. Note that NTK performs learning based on fixed features, which is shown to be limited [66]. In contrast, our algorithm allows hierarchical feature learning during the training process. In addition, our algorithm does not rely on discrete decision variables as in [6]: we use first-order methods with gradient and Hessian-inverse-vector product information with convergence guarantee and hence is more efficient in practice. For non-coreset reply methods SPR and MetaSP, we set the size of the method buffer as that in coreset-based methods. Especially for SPR, there are two memory buffers: the delayed buffer $D$ temporarily stocks the incoming data stream, and the purified buffer $P$ maintains the cleansed data. To satisfy the

Table 4: Experiment results on Food-101

| Methods | Balanced | | Imbalanced | | Label Noise | |
|---|---|---|---|---|---|---|
| | AVG ACC | FGT | AVG ACC | FGT | AVG ACC | FGT |
| K-means Features | 46.35±1.41 | 0.057±0.004 | 33.35±0.93 | 0.064±0.008 | 34.88±0.64 | 0.127±0.011 |
| K-means Embedding | 49.47±0.46 | 0.040±0.004 | 34.07±0.89 | 0.057±0.014 | 35.54±0.57 | 0.141±0.008 |
| Uniform | 46.50±0.13 | 0.048±0.001 | 34.61±0.16 | 0.060±0.007 | 36.69±1.13 | 0.123±0.007 |
| iCaRL | 48.12±0.78 | 0.035±0.001 | 36.83±1.06 | **0.032±0.001** | 39.22±0.30 | 0.084±0.010 |
| Grad Matching | 49.13±0.44 | **0.026±0.002** | 36.51±0.04 | 0.043±0.003 | 38.86±0.31 | 0.086±0.019 |
| SPR | 48.34±0.54 | 0.042±0.003 | 37.41±0.68 | 0.037±0.006 | 38.63±0.34 | 0.069±0.016 |
| MetaSP | 48.84±0.38 | 0.053±0.006 | 38.41±0.78 | 0.053±0.011 | 38.78±0.41 | 0.063±0.013 |
| Greedy Coreset | 49.19±0.91 | 0.030±0.008 | 38.78±0.01 | 0.038±0.001 | 38.61±0.18 | 0.096±0.004 |
| GCR | 47.03±1.17 | 0.048±0.004 | 36.99±0.30 | 0.042±0.005 | 35.78±1.08 | 0.126±0.013 |
| PBCS | 47.98±0.42 | 0.098±0.006 | 38.00±0.16 | 0.046±0.001 | 37.77±0.40 | 0.123±0.013 |
| OCS | 46.80±1.05 | 0.103±0.013 | 34.44±0.29 | 0.067±0.001 | 35.50±0.04 | 0.135±0.004 |
| BCSR | **51.30±0.62** | 0.039±0.008 | **39.05±0.99** | 0.044±0.001 | **39.63±0.22** | **0.060±0.071** |

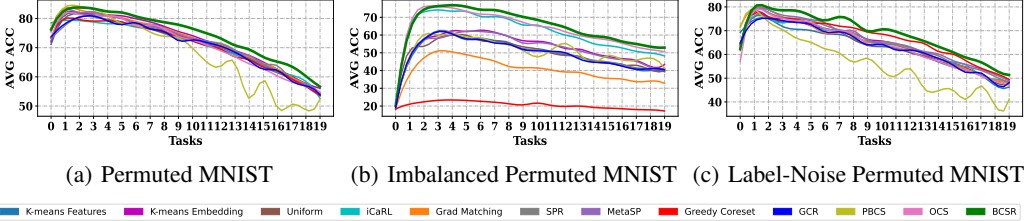

|  | (a) Permuted MNIST | (b) Imbalanced Permuted MNIST | (c) Label-Noise Permuted MNIST |

Figure 2: The average accuracy during the continual learning. After each task, the training model is tested on all encountered tasks. Due to the nature of forgetting, the average test performance of all the methods tends to decrease.

Table 5: Random initialization with (W) and without (WO) bilevel optimizers (BO)

| BO | Split CIFAR-100 | | Permuted MNIST | | Multiple Datasets | |
| | $A_T$ | $FGT_T$ | $A_T$ | $FGT_T$ | $A_T$ | $FGT_T$ |
|---|---|---|---|---|---|---|
| WO | 58.37±0.37 | 0.073±0.004 | 53.34±0.74 | 0.074±0.009 | 55.50±0.80 | 0.128±0.027 |
| W | **61.60±0.14** | **0.051±0.015** | **56.23±0.29** | **0.058±0.002** | **59.89±0.95** | **0.096±0.005** |

requirement of coreset experiments, we keep the size of $P$ and $D$ buffers the same. The detailed hyperparameter settings can be found in Appendix C.

## 5.2 Results

We report the results of average accuracy (AVG ACC) and forgetting (FGT) for all the algorithms on the balanced, imbalanced, and label-noise benchmarks, and the results are presented in Table 1, Table 2, Table 3, Table 4, and Table 9 (Appendix E) respectively. We have the following observations. (i) In the balanced setting, our method outperforms other baselines significantly in terms of AVG ACC. For example, compared with the best coreset selection methods, our BCSR shows $2.21\%$, $4.24\%$, $2.68\%$, $2.11\%$, and $1.7\%$ improvements in AVG ACC on five benchmarks, respectively. (ii) In the imbalanced and label-noise setting, our method also demonstrates relatively higher performance. For example, on imbalanced data of Multiple Datasets, BCSR is better than the best coreset selection algorithm by $4.65\%$ in average accuracy. An interesting observation is that Greedy Coreset [6] does not perform very well, especially in imbalanced and label-noise settings. The reason is that the inner optimization in Greedy Coreset is conducted by an NTK, where only a fixed feature is used but not the learned feature. (iii) From bilevel optimization point of view, our algorithm significantly reduces the running time of the vanilla greedy coreset selection method [6] by at least $58\%$ on continual learning benchmark datasets. For detailed comparison, please check Table 8 in Appendix D. (iv) In addition, the test AVG ACC (Figure 2 and Figure 4 in Appendix F) of the BCSR during the training process shows that our algorithm alleviates catastrophic forgetting and it is comparable to other best baselines. For example, BCSR enjoys the lowest forgetting for almost all datasets under the label-noise setting (except for Multiple Datasets in Table 2). For most experiments on other settings, BCSR achieves a forgetting performance that is comparable to the best methods with at most $1\%$ gap.

## 5.3 Ablation Studies

We conduct ablation studies to inspect the effectiveness of individual components in our proposed approach, including the effect of the bilevel optimizer, the smoothed top-$K$ regularizer, and the coreset size $K$ (Appendix J) respectively.

**Random initialization with and without bilevel optimizers.** Our new bilevel optimization algorithm (Algorithm 3) is of vital importance for finding a good distribution $w$ for selecting the coreset. To demonstrate this, we design comparative experiments: one experiment initializes $w$ randomly without updating it, and the other one adopts the same initialization strategy but updates $w$ by the proposed bilevel optimizer. Then we sample coresets based on $w$ from two methods respectively. As you can see, the random initialization method is equivalent to Uniform Sampling. We report the experimental results of average accuracy and forgetting on three balanced datasets in Table 5. It can be observed that the sampling strategy based on bilevel optimization significantly outperforms random sampling. In particular, the strategy with bilevel optimization shows $3.23\%$, $2.89\%$, and $4.39\%$ AVG ACC improvement on three benchmarks. Meanwhile, it also obtains $2.20\%$, $1.60\%$, and $3.20\%$ FGT reduction on three benchmarks.

Table 6: The impact of different values of $\lambda$

| | Split CIFAR-100 | | Permuted MNIST | | Multiple Datasets | |
|---|---|---|---|---|---|---|
| $\lambda$ | $A_T$ | $FGT_T$ | $A_T$ | $FGT_T$ | $A_T$ | $FGT_T$ |
| 0.00 | 60.43±0.15 | 0.064±0.005 | 54.20±1.53 | 0.116±0.030 | 55.71±0.32 | 0.055±0.003 |
| 0.01 | 60.56±2.05 | 0.074±0.016 | 55.09±2.94 | 0.097±0.020 | 55.39±0.95 | 0.065±0.012 |
| 0.10 | **61.04±0.53** | **0.063±0.007** | **57.20±0.58** | **0.064±0.010** | 55.69±0.40 | 0.057±0.002 |
| 1.00 | 59.43±1.20 | 0.072±0.008 | 55.43±1.53 | 0.108±0.019 | **58.18±0.77** | **0.046±0.006** |
| 5.00 | 58.80±1.58 | 0.084±0.017 | 53.13±2.74 | 0.120±0.021 | 56.67±0.78 | 0.051±0.007 |
| 10.00 | 59.53±0.42 | 0.075±0.012 | 53.54±2.02 | 0.118±0.029 | 55.57±1.05 | 0.060±0.010 |

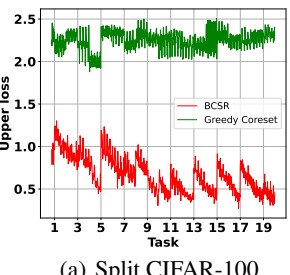

(a) Split CIFAR-100

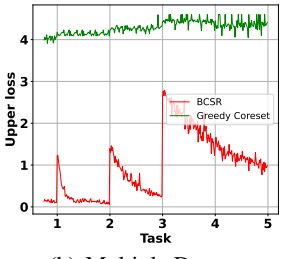

(b) Multiple Datasets

Figure 3: The upper loss of bilevel optimization. Each distinct spike means the arrival of a new task.

To verify the effectiveness of our bilevel optimizer, we compare the loss curve with Greedy Coreset that uses NTK. The result is presented in Figure 3. There are a number of stream mini-batches in each task. The bilevel optimizer trains over each mini-batch, where the upper loss is plotted in the figure. In the experiment, we plot the loss value for every 5 mini-batches. Within each task, the loss from BCSR gradually decreases with slight fluctuations, and it increases only when encountering a new task. In contrast, the loss value of the Greedy Coreset approach always stays large. It indicates that our bilevel optimizer is more effective than the Greedy Coreset approach.

**Effectiveness of the regularizer.** In our algorithm, the bilevel formulation has a smooth top-$K$ loss as a regularizer in the objective function to promote the probability distribution to have $K$ nonzero entries. The goal is to make sure that the summation of top-$K$ entries of the learned probability vector is large, which increases confidence in the coreset selection. The hyperparameter $\lambda$ is used to balance the cross-entropy loss and the regularization term. We explore the effects on the performance with different $\lambda$, and list the results in Table 6.

This ablation experiment is performed on our framework BCSR, and average accuracy and forgetting on three balanced benchmarks are reported. When $\lambda = 0.10$, BCSR can reach the best performance (The highest AVG ACC and lowest FGT) on Split CIFAR-100 and Permuted MNIST. While on Multiple Datasets, BCSR performs the best when $\lambda = 1.0$. This dataset contains more categories, and hence it is more challenging to select the appropriate representative samples. In this case, larger $\lambda$ would emphasize more on maximizing the top-$K$ probability and help select better representative samples. In addition, we observe that $\lambda$ set as a too large or too small value will damage the performance, which is in line with the observations in standard regularized empirical risk minimization problems such as overfitting and underfitting. Please refer to Appendix H for further analysis.

## 6 Theoretical Analysis

In this section, we provide a theoretical analysis for our proposed method. In particular, we establish convergence rate for our algorithm BCSR.

**Theorem 1.** *Suppose standard assumptions hold in bilevel optimization (see Assumptions 1, 2 and 3 in Appendix M). Choose parameters $\lambda, \alpha, \eta$ and $N$ such that $(1 + \lambda)(1 - \alpha\mu)^N(1 + \frac{8rL^2}{\eta\mu}) \leq 1 - \eta\mu$, where $r = \frac{C_Q^2}{(\frac{\rho M}{\mu} + L)^2}$ and $C_Q = \frac{Q\rho M\eta}{\mu} + \eta^2 Q^2 \rho M + \eta QL$. Furthermore, choose the stepsize $\beta$ such that $6\omega\beta^2 L^2 < \frac{1}{9}\eta\mu$ and $\beta \leq \frac{1}{4L_\phi}$, where the constant $\omega$ is given by Equation (16) in Appendix A.*

*Then, we have the following convergence result.*

$$\frac{1}{J}\sum_{j=0}^{J-1}\mathbb{E}\|G_j\|^2 \leq \mathcal{O}\Big(\frac{D_\phi}{\beta J} + \frac{1}{|\mathcal{B}|} + \frac{D_0}{\eta\mu J}\Big).$$

*where $G_j := \frac{1}{\beta}(w_j - \mathcal{P}_{\Delta^n}(w_j - \beta\nabla\phi(w_j)))$ denote the generalized projected gradient, $D_\phi := \phi(w_0) - \min_w \phi(w) > 0$, $D_0 = \|\theta_0^0 - \theta_0^*\|^2 + \|v_0^0 - v_0^*\|^2$, $L_\phi$ is the smoothness parameter of the total objective $\phi(w)$ whose form is proved in Appendix A.*

Theorem 1 provides a general convergence result for the proposed bilevel algorithm, which allows for a very flexible selection of subloop lengths $N$ and $Q$ as long as the inequality $(1+\lambda)(1-\alpha\mu)^N(1+\frac{8rL^2}{\eta\mu}) \leq 1 - \eta\mu$ holds given proper stepsizes $\lambda, \eta, \alpha$. For example, in most of our experiments, the choice of $N = 1$ and $Q = 3$ works the best. Then, in this case, we further specify the parameters in Theorem 1, and provide the following corollary.

**Corollary 1.** *Under the same setting as in Theorem 1, choose $N = 1, Q = 3$ and set $\lambda = \frac{\alpha\mu}{2}$, $\eta \leq \frac{\mu^2\alpha}{4608L^2}$ and $\alpha \leq \frac{1}{L}$. Then, to make sure an $\epsilon$-accurate stationary point, i.e., $\frac{1}{J}\sum_{j=0}^{J-1}\mathbb{E}\|G_j\|^2 \leq \epsilon^2$, the number of iterations is $\mathcal{O}(\epsilon^{-2})$, each using $|\mathcal{B}| = \mathcal{O}(\epsilon^{-2})$ of samples from the standard Gaussian distribution $\mathcal{N}(0,1)$.*

Corollary 1 shows that the proposed bilevel algorithm converges to an $\epsilon$-accurate stationary point using only $\mathcal{O}(\epsilon^{-2})$ iterations and $\mathcal{O}(\epsilon^{-2})$ samples drawn from $\mathcal{N}(0,1)$ per iteration. Note the the large batch size $|\mathcal{B}| = \mathcal{O}(\epsilon^{-2})$ is necessary here to guarantee a convergence rate of $\mathcal{O}(1/T)$, which matches the results in solving single-level constrained nonconvex problems [24]. In addition, it is computationally tractable because sampling from a known Gaussian distribution is easy and cheap.

## 7 Conclusion

In this paper, we advance the state-of-the-art bilevel coreset selection in continual learning. We first introduce a new bilevel formulation with smoothed top-$K$ regularization and then design an efficient bilevel optimizer as a solver. We conduct extensive experiments in continual learning benchmark datasets to demonstrate the effectiveness of our proposed approach. We also show that the bilevel optimizer can efficiently find $\epsilon$-stationary point with $O(1/\epsilon^4)$ computational complexity, which matches the best complexity of projected SGD for a single-level problem.

## Acknowledgments and Disclosure of Funding

We would like to thank the anonymous reviewers for their helpful comments. Jie Hao and Mingrui Liu are both supported by a grant from George Mason University. Computations were run on ARGO, a research computing cluster provided by the Office of Research Computing at George Mason University (URL: https://orc.gmu.edu).

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

# A Proof of Theorem 1

Let $\Delta^{(n)} := \{w : 0 \leq w_{(i)} \leq 1, \|w\|_1 = 1\}$ denote the constraint set of the upper-level problem. We first provide some important inequalities.

Recall that $v_j^q$ be the $q^{th}$ GD iterate in solving the linear system $\nabla_\theta^2 L(\theta_j^N, w_j) v = \sum_{i=1}^n \nabla \ell_i(\theta_j^N)$ at iteration $j$ via the following process:

$$v_j^{q+1} = (I - \eta \nabla_\theta^2 L(\theta_j^N, w_j)) v_j^q + \eta \sum_{i=1}^n \nabla \ell_i(\theta_j^N). \tag{6}$$

which, by telescoping Equation (6) over $q$ from 0 to $Q-1$, yields

$$v_j^Q = (I - \eta \nabla_\theta^2 L(\theta_j^N, w_j))^Q v_k^0 + \eta \sum_{q=0}^{Q-1} (I - \eta \nabla_\theta^2 L(\theta_j^N, w_j))^q \sum_{i=1}^n \nabla \ell_i(\theta_j^N). \tag{7}$$

Let $v_j^*$ be the solution of the linear system $\nabla_\theta^2 L(\theta_j^*, w_j) v = \sum_{i=1}^n \nabla \ell_i(\theta_j^*)$, and then we have

$$v_j^* = (I - \eta \nabla_\theta^2 L(\theta_j^*, w_j))^Q v_j^* + \eta \sum_{q=0}^{Q-1} (I - \eta \nabla_\theta^2 L(\theta_j^*, w_j))^q \sum_{i=1}^n \nabla \ell_i(\theta_j^*). \tag{8}$$

Combining Equation (6) and Equation (7), noting that $v_j^0 = v_{j-1}^Q$ and using Assumption 2 that $\|v_j^*\| \leq \|(\nabla_\theta^2 L(\theta_j^*, w_j))^{-1}\| \|\sum_{i=1}^n \nabla \ell_i(\theta_j^*)\| \leq \frac{M}{\mu}$, the difference between $v_j^Q$ and $v_j^*$ can be bounded as

$$\|v_j^Q - v_j^*\| \leq \|(I - \eta \nabla_\theta^2 L(\theta_j^N, w_j))^Q - (I - \eta \nabla_\theta^2 L(\theta_j^*, w_j))^Q\| \frac{M}{\mu} + (1 - \eta\mu)^Q \|v_{j-1}^Q - v_j^*\|$$

$$+ \eta M \left\| \sum_{q=0}^{Q-1} (I - \eta \nabla_\theta^2 L(\theta_j^N, w_j))^q - \sum_{q=0}^{Q-1} (I - \eta \nabla_\theta^2 L(\theta_j^*, w_j))^q \right\|$$

$$+ (1 - (1 - \eta\mu)^Q) \frac{L}{\mu} \|\theta_j^* - \theta_j^N\|. \tag{9}$$

We next bound $\Delta_q := \|(I - \eta \nabla_\theta^2 L(\theta_j^N, w_j))^q - (I - \eta \nabla_\theta^2 L(\theta_j^*, w_j))^q\|$ in Equation (9) as:

$$\Delta_q \overset{(i)}{\leq} (1 - \eta\mu) \Delta_{q-1} + (1 - \eta\mu)^{q-1} \eta\rho \|\theta_j^N - \theta_j^*\|. \tag{10}$$

which, by telescoping Equation (10) and in conjunction with Equation (9), yields

$$\|v_j^Q - v_j^*\| \leq Q(1 - \eta\mu)^{Q-1} \eta\rho \frac{M}{\mu} \|\theta_j^N - \theta_j^*\| + (1 - \eta\mu)^Q \|v_{j-1}^Q - v_j^*\|$$

$$+ \eta M \sum_{q=0}^{Q-1} q(1 - \eta\mu)^{q-1} \eta\rho \|\theta_j^N - \theta_j^*\| + (1 - (1 - \eta\mu)^Q) \frac{L}{\mu} \|\theta_j^* - \theta_j^N\| \tag{11}$$

$$\leq \frac{Q(1 - \eta\mu)^{Q-1} \rho M \eta}{\mu} \|\theta_j^N - \theta_j^*\| + (1 - \eta\mu)^Q \|v_{j-1}^Q - v_{j-1}^*\|$$

$$+ (1 - \eta\mu)^Q \|v_{j-1}^* - v_j^*\| + \frac{1 - (1 - \eta\mu)^Q (1 + \eta Q\mu)}{\mu^2} \rho M \|\theta_j^N - \theta_j^*\|$$

$$+ (1 - (1 - \eta\mu)^Q) \frac{L}{\mu} \|\theta_j^* - \theta_j^N\|$$

which, combined with $\|v_j^* - v_{j-1}^*\| \leq \left(\frac{L}{\mu} + \frac{M\rho}{\mu^2}\right)\left(\frac{L}{\mu} + 1\right) \|w_j - w_{j-1}\|$ and using the fact that $(1-x)^Q \geq 1 - xQ$ for $0 \leq x \leq 1$, yields

$$\mathbb{E}\|v_j^Q - v_j^*\|^2 \leq (1 - \eta\mu) \mathbb{E}\|v_{j-1}^Q - v_{j-1}^*\|^2 + \frac{4}{\eta\mu} C_Q^2 \|\theta_j^* - \theta_j^N\|^2$$

$$+ \frac{4}{\eta\mu}\left(\frac{L}{\mu} + \frac{M\rho}{\mu^2}\right)^2 \left(\frac{L}{\mu} + 1\right) \|w_j - w_{j-1}\|^2 \tag{12}$$

where the constant $C_Q := \frac{Q\rho M\eta}{\mu} + \eta^2 Q^2 \rho M + \eta Q L$.

The next step is to characterize the error induced by the lower-level updates on $\theta$.

Note that $\theta_j^* = \arg\min_\theta L(\theta, w_j)$. Using Assumptions 1 and 2, we have

$$\|\theta_j^N - \theta_j^*\|^2 \leq (1 - \alpha\mu)^N \|\theta_j^0 - \theta_j^*\|^2, \tag{13}$$

which, in conjunction with $\theta_j^0 = \theta_{j-1}^N$ and Lemma 2.2 in [25], yields

$$\mathbb{E}\|\theta_j^N - \theta_j^*\|^2 \leq (1 - \alpha\mu)^N (1 + \lambda) \mathbb{E}\|\theta_{j-1}^N - \theta_{j-1}^*\|^2$$
$$+ (1 - \alpha\mu)^N (1 + \frac{1}{\lambda}) \frac{L^2}{\mu^2} \mathbb{E}\|w_j - w_{j-1}\|^2. \tag{14}$$

Recall the definition that $r = \frac{C_Q^2}{(\frac{\rho M}{\mu} + L)^2}$. Then, combining Equation (12) and Equation (14) yields we can obtain

$$\left(1 + \frac{\rho^2 M^2}{L^2\mu^2}\right) \mathbb{E}\|\theta_j^N - \theta_j^*\|^2 + \mathbb{E}\|v_j^Q - v_j^*\|^2$$

$$\leq (1 + \lambda)(1 - \alpha\mu)^N \left(1 + \frac{\rho^2 M^2}{L^2\mu^2}\right) \left(1 + \frac{8rL^2}{\eta\mu}\right) \mathbb{E}\|\theta_{j-1}^N - \theta_{j-1}^*\|^2$$
$$+ (1 - \eta\mu) \mathbb{E}\|v_{j-1}^Q - v_{j-1}^*\|^2 + \omega \mathbb{E}\|w_j - w_{j-1}\|^2, \tag{15}$$

where the constant $\omega$ is given by

$$\omega := \left(1 + \frac{1}{\lambda}\right)(1 - \alpha\mu)^N \left(1 + \frac{\rho^2 M^2}{L^2\mu^2}\right) \frac{L^2}{\mu^2}$$
$$+ \frac{8}{\eta\mu} \frac{L^4}{\mu^2} \left(1 + \frac{\rho^2 M^2}{L^2\mu^2}\right) \left(\frac{4}{\mu^2} + r(1 - \alpha\mu)^N \left(1 + \frac{1}{\lambda}\right)\right). \tag{16}$$

Recall that we choose $(1 + \lambda)(1 - \alpha\mu)^N (1 + \frac{8rL^2}{\eta\mu}) \leq 1 - \eta\mu$. Then, we obtain from Equation (15) that

$$\left(1 + \frac{\rho^2 M^2}{L^2\mu^2}\right) \mathbb{E}\|\theta_j^N - \theta_j^*\|^2 + \mathbb{E}\|v_j^Q - v_j^*\|^2$$

$$\leq (1 - \eta\mu)\left(1 + \frac{\rho^2 M^2}{L^2\mu^2}\right) \mathbb{E}\|\theta_{j-1}^N - \theta_{j-1}^*\|^2$$
$$+ (1 - \eta\mu) \mathbb{E}\|v_{j-1}^Q - v_{j-1}^*\|^2 + \omega \mathbb{E}\|w_j - w_{j-1}\|^2. \tag{17}$$

Let $\delta_j := \left(1 + \frac{\rho^2 M^2}{L^2\mu^2}\right) \mathbb{E}\|\theta_j^N - \theta_j^*\|^2 + \mathbb{E}\|v_j^Q - v_j^*\|^2$. Then, incorporating the update that $w_j = \mathcal{P}_{\Delta^n}(w_{j-1} - \beta\varphi_{j-1})$ into Equation (17) yields

$$\delta_j \leq (1 - \eta\mu)\delta_{j-1} + 2\omega\beta^2 \mathbb{E}\left\|\underbrace{\frac{1}{\beta}(w_{j-1} - \mathcal{P}_{\Delta^n}(w_{j-1} - \beta\nabla\phi(w_{j-1})))}_{G_{j-1}}\right\|^2$$
$$+ 2\omega\beta^2 \|\varphi_{j-1} - \nabla\phi(w_{j-1})\|^2. \tag{18}$$

We next bound the last term in the above Equation (18). Based on the definition of the hypergradient estimate $\varphi_j = \nabla_w R(w, \delta; \mathcal{B}) - \nabla_w \nabla_\theta L(\theta_j^N, w_j) v_j^Q$, we have

$$\mathbb{E}\|\varphi_j - \nabla\phi(w_j)\|^2$$

$$\leq 3\mathbb{E}\|\nabla R(w_j, \delta; \mathcal{B}) - \nabla R(w_j, \delta)\|^2 + \frac{3\rho^2 M^2}{\mu^2} \mathbb{E}\|\theta_j^* - \theta_j^N\|^2 + 3L^2 \mathbb{E}\|v_j^* - v_j^Q\|^2$$

$$\overset{(i)}{=} \frac{3}{|\mathcal{B}|^2} \sum_{\widetilde{z} \in \mathcal{B}} \|\nabla R(w_j, \delta; \widetilde{z}) - \nabla R(w, \delta)\|^2 + 3L^2 \left(1 + \frac{\rho^2 M^2}{\mu^2 L^2}\right) \mathbb{E}\|\theta_j^* - \theta_j^N\|^2 + 3L^2 \mathbb{E}\|v_j^* - v_j^Q\|^2$$

$$\overset{(ii)}{\leq} \frac{6K}{|\mathcal{B}|} + 3L^2 \left(1 + \frac{\rho^2 M^2}{\mu^2 L^2}\right) \mathbb{E}\|\theta_j^* - \theta_j^N\|^2 + 3L^2 \mathbb{E}\|v_j^* - v_j^Q\|^2 = \frac{6K}{|\mathcal{B}|} + 3L^2 \delta_j, \tag{19}$$

where $(i)$ follows because $\nabla R(w_j, \delta; \widetilde{z})$ is an unbiased estimate of $\nabla R(w, \delta)$ and $(ii)$ follows from Proposition 1. Substituting Equation (19) into Equation (18) yields

$$\delta_j \leq (1 - \eta\mu + 6\omega\beta^2 L^2)\delta_{j-1} + \frac{12\omega K\beta^2}{|\mathcal{B}|} + 2\omega\beta^2 \mathbb{E}\|G_{j-1}\|^2. \tag{20}$$

Let $\tau := 1 - \eta\mu + 6\omega\beta^2 L^2$. Then, telescoping Equation (20) yields

$$\delta_j \leq \tau^j \delta_0 + \frac{12\omega K\beta^2}{(1-\tau)|\mathcal{B}|} + 2\omega\beta^2 \sum_{t=0}^{j-1} \tau^t \mathbb{E}\|G_{j-1-t}\|^2. \tag{21}$$

Based on Equation (21), we are ready to provide the final convergence result. First, based on the Lipschitz continuity in Assumption 1, Assumption 2 and Assumption 3, we have

$$\|\nabla\phi(w_1) - \nabla\phi(w_2)\| \leq L_\phi \|w_1 - w_2\|,$$

where the constant $L_\phi = \frac{\sqrt{Kn}}{\delta} + \frac{L^2 + \rho M^2}{\mu} + \frac{2\rho LM + L^3}{\mu^2} + \frac{\rho L^2 M}{\mu^3}$ is the smoothness parameter. Then, this inequality further implies

$$\phi(w_{j+1}) \leq \phi(w_j) + \langle\nabla\phi(w_j), w_{j+1} - w_j\rangle + \frac{L_\phi}{2}\|w_{j+1} - w_j\|^2$$

$$\leq \phi(w_j) + \frac{1}{\beta}\langle\beta\varphi_j, \mathcal{P}_{\Delta^n}(w_j - \beta\varphi_j) - w_j\rangle + \langle\nabla\phi(w_j) - \varphi_j, \mathcal{P}_{\Delta^n}(w_j - \beta\varphi_j) - w_j\rangle$$

$$+ \frac{L_\phi}{2}\|w_{j+1} - w_j\|^2. \tag{22}$$

To analyze the second term at the right hand side of the above Equation (22), we note that

$$-\langle\beta\varphi_j, \mathcal{P}_{\Delta^n}(w_j - \beta\varphi_j) - w_j\rangle$$

$$= \langle w_j - \beta\varphi_j - \mathcal{P}_{\Delta^n}(w_j - \beta\varphi_j), \mathcal{P}_{\Delta^n}(w_j - \beta\varphi_j) - w_j\rangle + \|\mathcal{P}_{\Delta^n}(w_j - \beta\varphi_j) - w_j\|^2,$$

which, in conjunction with the property of projection on convex set that $\langle x - \mathcal{P}_{\Delta^n}(x), y - \mathcal{P}_{\Delta^n}(x)\rangle \leq 0$ for any $y \in \mathcal{S}$ and the fact that $w_j = \mathcal{P}_{\Delta^n}(w_{j-1} - \beta\varphi_{j-1}) \in \mathcal{S}$, yields

$$-\langle\beta\varphi_j, \mathcal{P}_{\Delta^n}(w_j - \beta\varphi_j) - w_j\rangle \geq \|\mathcal{P}_{\Delta^n}(w_j - \beta\varphi_j) - w_j\|^2 \geq 0. \tag{23}$$

Then, substituting Equation (23) into Equation (22) yields

$$\phi(w_{j+1}) \leq \phi(w_j) + \langle\nabla\phi(w_j) - \varphi_j, \mathcal{P}_{\Delta^n}(w_j - \beta\varphi_j) - w_j\rangle + \frac{L_\phi}{2}\|w_{j+1} - w_j\|^2$$

$$\leq \phi(w_j) - \frac{\beta}{2}\|\widehat{G}_j\|^2 + \frac{\beta}{2}\|\varphi_j - \nabla\phi(w_j)\|^2 + \frac{L_\phi\beta^2}{2}\|\widehat{G}_j\|^2$$

$$\leq \phi(w_j) - \left(\frac{\beta}{4} - \frac{L_\phi\beta^2}{4}\right)\|G_j\|^2 + \left(\beta - \frac{L_\phi\beta^2}{2}\right)\|\varphi_j - \nabla\phi(w_j)\|^2, \tag{24}$$

where we use the notation that $\widehat{G}_j = \frac{1}{\beta}\left(w_j - \mathcal{P}_{\Delta^n}(w_j - \beta\varphi_j)\right)$, and the non-expansive property of projection. Then, taking the expectation and incorporating Equation (19) into Equation (24), we have

$$\mathbb{E}\phi(w_{j+1}) \leq \mathbb{E}\phi(w_j) - \left(\frac{\beta}{4} - \frac{L_\phi\beta^2}{4}\right)\mathbb{E}\|G_j\|^2 + \left(\beta - \frac{L_\phi\beta^2}{2}\right)\mathbb{E}\|\varphi_j - \nabla\phi(w_j)\|^2$$

$$\leq \mathbb{E}\phi(w_j) - \left(\frac{\beta}{4} - \frac{L_\phi\beta^2}{4}\right)\mathbb{E}\|G_j\|^2 + \left(\beta - \frac{L_\phi\beta^2}{2}\right)\left(\frac{6K}{|\mathcal{B}|} + 3L^2\delta_j\right). \tag{25}$$

Then, substituting Equation (21) into the above Equation (25) yields

$$\mathbb{E}\phi(w_{j+1}) \leq \mathbb{E}\phi(w_j) - \left(\frac{\beta}{4} - \frac{L_\phi\beta^2}{4}\right)\mathbb{E}\|G_j\|^2 + \left(\beta - \frac{L_\phi\beta^2}{2}\right)\frac{6K}{|\mathcal{B}|}$$

$$+ 3L^2\left(\beta - \frac{L_\phi\beta^2}{2}\right)\left(\tau^j\delta_0 + \frac{12\omega K\beta^2}{(1-\tau)|\mathcal{B}|} + 2\omega\beta^2 \sum_{t=0}^{j-1} \tau^t \mathbb{E}\|G_{j-1-t}\|^2\right)$$

$$\leq \mathbb{E}\phi(w_j) - \left(\frac{\beta}{4} - \frac{L_\phi\beta^2}{4}\right)\mathbb{E}\|G_j\|^2 + \left(1 + \frac{6\omega\beta^2 L^2}{1-\tau}\right)\left(\beta - \frac{L_\phi\beta^2}{2}\right)\frac{6K}{|\mathcal{B}|}$$

$$+ 3L^2\left(\beta - \frac{L_\phi\beta^2}{2}\right)\tau^j\delta_0 + 6\omega\beta^2 L^2\left(\beta - \frac{L_\phi\beta^2}{2}\right)\sum_{t=0}^{j-1} \tau^t \mathbb{E}\|G_{j-1-t}\|^2,$$

which, by taking the telescoping over $j$ from 0 to $J-1$, yields

$$\frac{1}{J}\sum_{j=0}^{J-1}\Big(\frac{1}{4}-\frac{L_\phi\beta}{4}\Big)\mathbb{E}\|G_j\|^2$$

$$\leq\frac{\phi(w_0)-\min_w\phi(w)}{\beta J}+\Big(1+\frac{6\omega\beta^2 L^2}{1-\tau}\Big)\Big(1-\frac{L_\phi\beta}{2}\Big)\frac{6K}{|\mathcal{B}|}$$

$$+\frac{3L^2\Big(1-\frac{L_\phi\beta}{2}\Big)\delta_0}{(1-\tau)J}+6\omega\beta^2 L^2\Big(1-\frac{L_\phi\beta}{2}\Big)\frac{1}{J}\sum_{j=0}^{J-1}\sum_{t=0}^{j-1}\tau^t\mathbb{E}\|G_{j-1-t}\|^2$$

$$\leq\frac{\phi(w_0)-\min_w\phi(w)}{\beta J}+\Big(1+\frac{6\omega\beta^2 L^2}{1-\tau}\Big)\Big(1-\frac{L_\phi\beta}{2}\Big)\frac{6K}{|\mathcal{B}|}$$

$$+\frac{3L^2\Big(1-\frac{L_\phi\beta}{2}\Big)\delta_0}{(1-\tau)J}+6\omega\beta^2 L^2\Big(1-\frac{L_\phi\beta}{2}\Big)\frac{1}{(1-\tau)J}\sum_{j=0}^{J-1}\mathbb{E}\|G_j\|^2.$$

Rearranging the above inequality, we have

$$\frac{1}{J}\sum_{j=0}^{J-1}\Big(\frac{1}{4}-\frac{L_\phi\beta}{4}-\frac{6\omega\beta^2 L^2}{1-\tau}\Big(1-\frac{L_\phi\beta}{2}\Big)\Big)\mathbb{E}\|G_j\|^2$$

$$\leq\frac{\phi(w_0)-\min_w\phi(w)}{\beta J}+\Big(1+\frac{6\omega\beta^2 L^2}{1-\tau}\Big)\Big(1-\frac{L_\phi\beta}{2}\Big)\frac{6K}{|\mathcal{B}|}+\frac{3L^2\Big(1-\frac{L_\phi\beta}{2}\Big)\delta_0}{(1-\tau)J}. \qquad (26)$$

Recalling the definition that $\tau=1-\eta\mu+6\omega\beta^2 L^2$, and noting that we choose the stepsize $\beta$ such that $6\omega\beta^2 L^2<\frac{1}{9}\eta\mu$ and $\beta\leq\frac{1}{4L_\phi}$, we can simplify Equation (26) as

$$\frac{1}{16J}\sum_{j=0}^{J-1}\mathbb{E}\|G_j\|^2\leq\frac{\phi(w_0)-\min_w\phi(w)}{\beta J}+\frac{27K}{4|\mathcal{B}|}+\frac{27L^2\delta_0}{8\eta\mu J}.$$

From the gradient descent based updates, we have $\delta_0=\big(1+\frac{\rho^2 M^2}{L^2\mu^2}\big)\mathbb{E}\|\theta_0^N-\theta_0^*\|^2+\mathbb{E}\|v_0^Q-v_0^*\|^2\leq\big(1+\frac{\rho^2 M^2}{L^2\mu^2}\big)\|\theta_0^0-\theta_0^*\|^2+\|v_0^0-v_0^*\|^2<+\infty$. Then, the proof is complete.

## B  Proof of Corollary 1

Based on the definition of $C_Q=\frac{Q\rho M\eta}{\mu}+\eta^2 Q^2\rho M+\eta QL$ and noting that $\eta\leq\frac{1}{L}\leq\frac{1}{\mu}$ and $Q=3$, we have $C_Q\leq 12\eta\big(\frac{\rho M}{\mu}+L\big)$, which, combined with the definition that $r=\frac{C_Q^2}{(\frac{\rho M}{\mu}+L)^2}$, yields $r\leq 144\eta^2$. Then, based on the choice of $N=1$ and $\lambda=\frac{\alpha\mu}{2}$, we have

$$(1+\lambda)(1-\alpha\mu)^N\Big(1+\frac{8rL^2}{\eta\mu}\Big)\leq\Big(1-\frac{\alpha\mu}{2}\Big)\Big(1+\frac{1152\eta L^2}{\mu}\Big)\leq 1-\frac{\alpha\mu}{2}+\frac{1152\eta L^2}{\mu},$$

which, in conjunction with $\eta\leq\frac{\mu^2}{4608L^2}\alpha$ and $\alpha\leq\frac{1}{L}$, yields

$$(1+\lambda)(1-\alpha\mu)^N\Big(1+\frac{8rL^2}{\eta\mu}\Big)\leq 1-\frac{1}{4}\alpha\mu\leq 1-\eta\mu. \qquad (27)$$

This implies that the inequality $(1+\lambda)(1-\alpha\mu)^N(1+\frac{8rL^2}{\eta\mu})\leq 1-\eta\mu$ required by Theorem 1 is satisfied. Then, treating $\mu,\eta,L,\rho,M,\alpha,\beta,K,\|\theta_0^0-\theta_0^*\|^2$ and $\|v_0^0-v_0^*\|^2$ as constants independent of the total number $J$ of iterations, we have

$$\frac{1}{J}\sum_{j=0}^{J-1}\mathbb{E}\|G_j\|^2\leq\mathcal{O}\Big(\frac{1}{J}+\frac{1}{|\mathcal{B}|}\Big). \qquad (28)$$

Then, to ensure an $\epsilon$-accurate stationary point, the number of iterations is $\epsilon^{-2}$ with a batch size $|\mathcal{B}|=\mathcal{O}(\epsilon^{-2})$.

## C   Experiment Hyperparameters

The experimemtal hyperparameters are list in Table 7, including the memory size: $|\mathcal{M}|$, coreset size: $|\mathcal{S}|$, stream batch size: $|\mathcal{B}_t|$, learning rate for $M_{tr}$: $lr_t$, learning rate for $M_{cs}$: $lr_p$, learning rate for sample weights: $lr_w$, regularization efficient: $\lambda$, epochs for training model: $E$, outer loops in bilevel optimization: $J$, inner loops in bilevel optimization: $N$, loops for estimating the Hessian-inverse-vector product: $Q$, a factor of Gaussian noise: $\delta$.

Table 7: Hyperparameters settings in experiments.

| Hyperparameters | Permuted MNIST | Split CIFAR-100 | Split Tiny-Imagenet | Multiple Datasets | Split Food-101 |
|---|---|---|---|---|---|
| $|\mathcal{M}|$ | 200 | 100 | 200 | 83 | 100 |
| $|\mathcal{S}|$ | 10 | 10 | 20 | 10 | 10 |
| $|\mathcal{B}_t|$ | 50 | 50 | 100 | 50 | 50 |
| $lr_t$ | 0.005 | 0.15 | 0.20 | 0.1 | 0.15 |
| $lr_p$ | 5.0 | 5.0 | 10 | 5.0 | 10 |
| $lr_w$ | 5.0 | 5.0 | 10 | 5.0 | 10 |
| $\lambda$ | 0.1 | 0.1 | 0.1 | 0.1 | 0.1 |
| $E$ | 1 | 1 | 1 | 1 | 1 |
| $J$ | 5 | 10 | 5 | 5 | 5 |
| $N$ | 1 | 1 | 1 | 1 | 1 |
| $Q$ | 3 | 3 | 3 | 3 | 3 |
| $\delta$ | $1e-3$ | $1e-3$ | $1e-3$ | $1e-3$ | $1e-3$ |

## D   Running Time Comparison

We evaluate the running time (wall-clock time) of baseline methods on Split Tiny-ImageNet and Split Food-101 in Table 8. All the training sections keep the same among these algorithms. Since there is no advanced coreset selection procedures in K-means Features, K-means Embedding, Uniform sampling, iCaRL, and Grad Matching, their running costs are pretty low but with worse performance (i.e., accuracy and forgetting) compared with our approach BCSR. PBCS and GCR, SPR and MetaSP don't involve bilevel formulations, they cannot guarantee accurate coreset sampling though with lower time cost against BCSR. Compared with OCS, BCSR takes 23% and 35% running time reduction on Tiny-ImageNet and Split Food-101, respectively. Because similarity computing based on data pairs is computationally expensive for OCS. Greedy Coreset takes much more time than BCSR due to the usage of NTK.

Table 8: Running time.

| Methods | Time (hours) | |
|---|---|---|
| | Tiny-ImageNet | Split Food-101 |
| K-means Features | 0.15 | 0.02 |
| K-means Embedding | 0.41 | 0.04 |
| Uniform | 0.13 | 0.03 |
| iCaRL | 0.41 | 0.04 |
| Grad Matching | 0.57 | 0.06 |
| SPR | 0.77 | 0.11 |
| MetaSP | 0.86 | 0.13 |
| PBCS | 1.13 | 0.15 |
| GCR | 0.91 | 0.13 |
| Greedy Coreset | 6.26 | 0.83 |
| OCS | 3.45 | 0.40 |
| BCSR | 2.65 | 0.26 |

## E   Experiments on Permuted-MNIST

The experiment results on Permuted-MNIST is presented in this section. BCSR outperforms other baselines significantly in AVG ACC on all the data settings, while showing relatively low forgetting.

Table 9: Experiment results on Permuted MNIST

| Methods | Balanced | | Imbalanced | | Label Noise | |
|---|---|---|---|---|---|---|
| | $A_T$ | $FGT_T$ | $A_T$ | $FGT_T$ | $A_T$ | $FGT_T$ |
| K-means Features | 54.30±0.93 | 0.064±0.012 | 39.46±0.11 | 0.023±0.003 | 46.71±0.64 | 0.094±0.011 |
| K-means Embedding | 54.78±1.83 | 0.056±0.009 | 41.97±2.02 | 0.013±0.005 | 47.36±1.58 | 0.082±0.015 |
| Uniform | 53.74±0.35 | 0.073±0.006 | 38.49±1.89 | 0.025±0.008 | 46.35±1.72 | 0.091±0.014 |
| iCaRL | 52.62±0.01 | 0.076±0.001 | 48.64±0.58 | 0.022±0.001 | 48.14±0.83 | 0.082±0.007 |
| Grad Matching | 54.76±1.61 | 0.065±0.011 | 33.67±1.24 | 0.028±0.003 | 48.24±0.42 | 0.089±0.009 |
| SPR | 54.24±0.45 | 0.068±0.019 | 40.79±0.83 | 0.031±0.003 | 48.23±0.75 | 0.067±0.004 |
| MetaSP | 54.63±0.31 | 0.059±0.006 | 41.32±0.94 | 0.025±0.006 | 48.84±0.77 | 0.061±0.007 |
| Greedy Coreset | 54.10±0.81 | 0.051±0.007 | 26.68±13.39 | 0.033±0.014 | 49.45±1.73 | 0.078±0.009 |
| GCR | 54.53±0.64 | 0.067±0.021 | 40.63±0.50 | 0.032±0.005 | 48.64±0.75 | 0.074±0.014 |
| PBCS | 51.61±1.14 | 0.144±0.021 | 41.14±0.23 | 0.119±0.007 | 39.74±1.98 | 0.178±0.001 |
| OCS | 54.37±0.34 | **0.026±0.001** | 50.19±0.47 | 0.020±0.005 | 48.08±1.44 | **0.046±0.003** |
| BCSR | **56.23±0.29** | 0.058±0.002 | **52.52±0.43** | **0.010±0.002** | **50.82±3.03** | 0.056±0.017 |

# F Evolution of Average Accuracy

We present the learning process of different methods on Multiple Datasets in Figure 4. There are five tasks, where the average accuracy is tested after training each task. BCSR shows better performance than other baselines in different settings, including Balanced, Imbalanced, and label-noise.

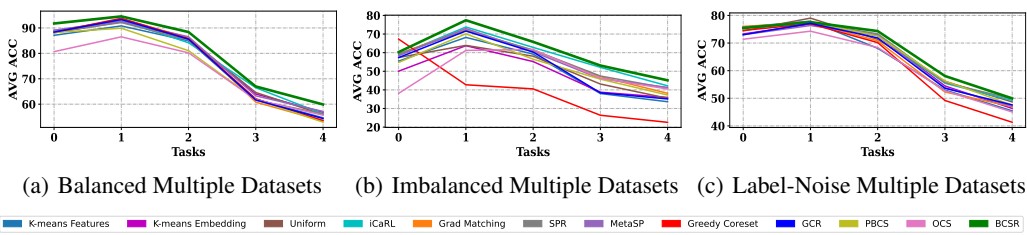

(a) Balanced Multiple Datasets  (b) Imbalanced Multiple Datasets  (c) Label-Noise Multiple Datasets

Figure 4: Evolution of average accuracy during the continual learning process for Multiple Datasets.

# G Possibility Distribution of Candidate Coreset

To explore the effect of top-$K$ regularizer, we observe the possibility distribution of sample weights after each bilevel optimization, where sample weights of candidate coresets are initialized uniformly. The novel top-$K$ regularizer makes sure that the summation of the top-$K$ entries of the learned probability vector is large, such that we can confidently choose a coreset with size $K$. We show the weight distribution after optimization in Figure. 5, where you can find that top-$K$ weights are much higher than others (with a margin $2\%$), which easily distinguishes the top-$K$ core samples from candidates.

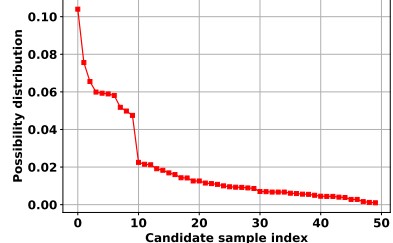

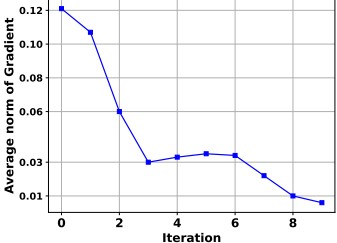

Figure 5: Possibility distribution of candidate coreset for one mini-batch stream data.

Figure 6: The hypergradients in an outer-loop.

## H The Effect of Top-$K$ Regularizer

To further analyze the effects of top-$K$ regularizer, we conduct the ablation study with different values of regularizer coefficient $\lambda$ on balanced Split CIFAR-100. The performance results with different $\lambda$ are shown in Table 10 and the corresponding average top-$K$ summations of coreset weights are in Table 11. In our experiment, there are 50 candidate samples in each mini-batch data, and the summation of 50 coreset weights is equal to 1.00. Top-$K$ summation of weights increases as $\lambda$ increases, which imposes higher probabilities on top-$K$ entries and lower probabilities on the rest candidates. The best performance is achieved when $\lambda = 0.1$, which means $\lambda$ balances the trade-off between the loss function and regularizer strength: if $\lambda$ is too large, the algorithm primarily focuses on choosing the important samples instead of updating the model parameter, and vice versa.

Table 10: Ablation study for the regularize coefficient $\lambda$.

| Measure | $\lambda$=0.01 | $\lambda$=0.05 | $\lambda$=0.10 | $\lambda$=0.50 | $\lambda$=1.00 |
|---------|---------|---------|---------|---------|---------|
| $A_T$ | 59.37±0.35 | 60.23±0.43 | **61.60±0.14** | 59.42±1.45 | 58.89±1.64 |
| $FGT_T$ | 0.095±0.098 | 0.074±0.054 | **0.051±0.015** | 0.138±0.075 | 0.128±0.076 |

Table 11: Top-$K$ summation/Total summation of coreset weights ($K = 10$).

| Measure | $\lambda$=0.01 | $\lambda$=0.05 | $\lambda$=0.10 | $\lambda$=0.50 | $\lambda$=1.00 |
|---------|---------|---------|---------|---------|---------|
| Top-$K$ Sum/Total Sum | 0.41/1.00 | 0.56/1.00 | 0.63/1.00 | 0.73/1.00 | 0.84/1.00 |

## I Hypergradient Evolution

To demonstrate the efficiency of bilevel optimization, we illustrate the evolution of hypergradients for a bilevel optimization on Split CIFAR-100 in Figure. 6. We observe the average norm of hypergradient reduces from $10^{-1}$ to less than $10^{-2}$ in each round of coreset selection with loops equal to 10. The hypergradient curves show that our designed bilevel optimization provides both theoretical and practical convergence guarantees.

Table 12: The effect of coreset size

| Methods | $K$=10 $A_T$ | $K$=20 $A_T$ | $K$=40 $A_T$ |
|---------|---------|---------|---------|
| Uniform | 58.99±0.54 | 53.57±2.93 | 53.03±1.97 |
| Greedy Coreset | 59.39±0.16 | 56.81±3.32 | 56.09±0.42 |
| PBCS | 55.64±2.26 | 49.84±1.76 | 40.95±0.32 |
| OCS | 52.57±0.37 | 54.87±0.58 | 56.46±0.07 |
| BCSR | **61.60±0.14** | **59.06±1.15** | **56.58±0.21** |

## J The Effect of Coreset Size $K$

Coreset size $K$ also plays an important role in the experiment. We compare our BCSR with other coreset-based algorithms on different $K$. Note that coreset is selected from the current stream mini-batch $\mathcal{B}_t$, so the coreset size $K$ satisfies $K \leq |\mathcal{B}_t|$. In the main experiment results, $K = 10$ is fixed in all the algorithms for a fair comparison. Here, we set $K = 10, 20, 40$ to conduct the continual learning experiments, respectively. The result is represented in Table 12. Note that all the results presented here are based on balanced Split CIFAR-100.

We mainly compare with the other four methods, including Uniform Sampling and three coreset-based methods, Greedy Coreset, PBCS, and OCS. We can observe that the performance of almost all methods becomes worse when coreset size is large. The reason is that if coreset size is larger and closer to $\mathcal{B}_t$, more redundant or noisy data are selected from the current stream mini-batch, and the coreset would not be representative anymore. In contrast, the smaller coreset could reduce the probability that redundant data are selected. Compared with other methods, BCSR shows the best performance (both accuracy and forgetting) and better robustness on different $K$.

# K  The Effect of Inner Loops $N$ and Hessian-inverse-vector Product Loops $Q$

We conduct ablation studies to explore the sensitivity of hyperparameters ($N$ and $Q$) on Split CIFAR-100. The results are presented in Tables 13 ($N$) and Table 14 ($Q$). The model performance remains relatively stable when increasing inner loops ($N$) while fixing $Q$. But too large $N$ ($N \geq 15$) leads to performance degradation due to overfitting. The $Q$ loops show similar properties that a few $Q$ loops (e.g., $Q = 3$) are enough to approximate the Hessian-inverse-vector product. Too small $Q$ and too large $Q$ will hurt the performance due to possible underfitting (e.g., $Q = 1$) and overfitting (e.g., $Q = 20$).

Table 13: Ablation study for the inner loops (N) with fixed $Q = 3$.

| Measure | $N$=1 | $N$=5 | $N$=10 | $N$=15 | $N$=20 |
|---|---|---|---|---|---|
| $A_T$ | 61.60±0.14 | **61.75±0.11** | 61.64±0.15 | 60.77±0.32 | 59.20±0.41 |
| $FGT_T$ | 0.051±0.015 | **0.047±0.013** | 0.063±0.017 | 0.074±0.021 | 0.079±0.035 |

Table 14: Ablation study for the loops $Q$ with fixed $N = 1$.

| Measure | $Q$=1 | $Q$=3 | $Q$=5 | $Q$=10 | $Q$=20 |
|---|---|---|---|---|---|
| $A_T$ | 52.14±1.53 | **61.60±0.14** | 61.57±0.15 | 58.42±0.53 | 57.80±1.31 |
| $FGT_T$ | 0.123±0.038 | **0.051±0.015** | 0.064±0.012 | 0.173±0.045 | 0.162±0.041 |

# L  Datasets

- **Spit CIFAR-100**. Balanced split CIFAR-100 is based on the original CIFAR-100 and is split into 20 tasks, each consisting of 5 disjoint classes. The imbalanced setting and label noise are also applied to this dataset to make the task more challenging. We follow [13] to transform the original dataset to imbalanced long-tailed CIFAR-100. In the label-noise setting, we randomly select 20% data in each task and randomly change their labels to an arbitrary label of 10 classes.

- **Permuted MNIST**. Balanced MNIST is a handwritten digits dataset [43] containing 20 tasks, where each task applies a fixed random permutation to the image pixels. For the imbalanced setting, we randomly select 8 classes over 10 and sample 10% of the selected classes for training. We also conduct the experiments on the label noise scenario, where symmetric label noise with 20% noise rate is imposed on the data. In particular, to make the problem setting more challenging, each task only retains 3000 training data randomly sampled from the original data.

- **Multiple Datasets**. This dataset [78] contains a couple of totally different datasets, including MNIST [43], fashion-MNIST [75], NotMNIST [8], Traffic Sign [69] and SVHN [54]. There are 5 tasks, and each task is constructed by randomly selecting 1000 training samples from a different dataset. The procedure of creating the dataset in the imbalanced and label-noise settings is the same as that in Split CIFAR-100.

- **Split Tiny-ImageNet**. This dataset [42] contains 100000 images of 200 classes (500 for each class) downsized to $64 \times 64$ colored images. Each class has 500 training images, 50 validation images, and 50 test images. We construct the task sequences by splitting data into 20 tasks, where each task consists of 10 disjoint classes.

- **Split Food-101**. This dataset [7] is a challenging data set of 101 food categories, with 101'000 images. All the images are resized to $64 \times 64$ pixels to be fed into models. To build continual learning tasks easily, We discard the last category and split data into 20 tasks, with 5 categories within each task. Other data settings, including imbalanced and lable-noise, are the same as Split CIFAR-100.

# M  Assumptions and Properties

We first provide standard definitions and assumptions for the convergence rate analysis of bilevel optimization [25, 36]. For notational convenience, we define $\ell(\theta) := \sum_{i=1}^{n} \ell_i(\theta)$.

**Definition 1.** *A mapping $f$ is $L_f$-Lipschitz continuous if for $\forall z, z'$, $\|f(z) - f(z')\| \leq L_f \|z - z'\|$.*

**Assumption 1.** *The lower-level function $L(\theta, w)$ is $\mu$-strongly-convex w.r.t. $\theta$.*

Assumption 1 is a necessary geometric assumption in analyzing the convergence rate of bilevel optimization algorithms, as also widely adopted existing theories in [25, 37, 32]. We also note that this condition is satisfied for overparameterized neural networks [82]. The following assumption imposes some Lipschitz continuity conditions on the upper- and lower-level objective functions.

**Assumption 2.** *The gradients $\nabla_\theta L(\theta, w)$, $\nabla_w L(\theta, w)$ and $\ell(\theta)$ are $L$-Lipschitz continuous w.r.t. $\theta$ and $w$. In addition, the gradient norm $\|\nabla \ell(\theta^*(w))\| \leq M$.*

Note that we do not impose any conditions on the regularization function $R(w, \delta)$. The following assumption imposes the Lipschitz continuity on the second-order derivatives of the lower-level functions.

**Assumption 3.** *The second-order derivatives $\nabla_w \nabla_\theta L(\theta, w)$ and $\nabla_\theta^2 L(w, \theta)$ are $\rho$-Lipschitz continuous.*

Then, we use the following proposition to characterize the properties of the smoothed top-$K$ regularizer $R(w, \delta)$, based on the results in [23].

**Proposition 1.** *The smoothed regularizer $R(w, \delta)$ and its sampled version $R(w, \delta; \widetilde{z})$ satisfy the following two important properties: (i) The gradient $\nabla R(w, \delta)$ exists and is $\frac{\sqrt{Kn}}{\delta}$-Lipschitz continuous; (ii) The gradient norm $\|R(w, \delta; \widetilde{z})\|$ is bounded by $\sqrt{K}$ for any sample $\widetilde{z}$.*

Proposition 1 shows that the regularizer $R(w, \delta)$ is smooth and its stochastic version $R(w, \delta; \widetilde{z})$ is bounded. These two properties are important to guarantee the non-asymptotic convergence of our proposed bilevel method.

