# OpenReview forum: "Bilevel Coreset Selection in Continual Learning: A New Formulation and Algorithm"
_NeurIPS.cc/2023/Conference — NeurIPS 2023 poster_

### Official Review · Reviewer_b44u · 2023-06-30

**Soundness:** 3 good
**Presentation:** 2 fair
**Contribution:** 3 good
**Rating:** 6
**Confidence:** 3

**Summary:**

Past works have limitations in terms of scalability, formulation approximation, or performance. This paper offers an efficient coreset selection problem with provable theoretical guarantees. That is, the authors solve a bilevel optimization on a probability distribution over the dataset with loss minimization on the selected dataset that is guided into a low-dimensional manifold via a smoothed top-K loss as a regularizer on the probability distribution.

**Strengths:**

The approach is simple, yet well-motivated by the limitations of previous works.
The writing addresses the limitations well and is easy to follow.


**Weaknesses:**

(1) Further analysis into the top-K loss and its causal effects would aid in understanding the mechanics of probability distribution being regularized into a low-dimensional manifold.

(2) comparisons with state-of-the-art non-coreset replay methods (e.g., using stability plasticity scores [1], contrastive representation based selection [2]), as it would be meaningful to see the prospect of this direction.

(3) [minor] main illustration figure 1 could include more technical information (embedding some important eqns for example) or include an additional figure to guide the reader better.



[1] Sun et al, Exploring Example Influence in Continual Learning, NeurIPS 2022
[2] Kim et al, Continual Learning on Noisy Data Streams via Self-Purified Replay, ICCV 2021

**Questions:**

if Weaknessses (1) could be further analyzed and shown, as well as comparison with (2) other state of the art rehearsal baselines comparison would be informative.

**Limitations:**

Not mentioned in the paper.

---

> ### Author Rebuttal · Authors · 2023-08-08
>
> Dear Reviewer b44u,
>
> Thanks for your reviews. We have addressed your concerns below.
>
> **Q1: Further analysis into the top-K loss and its causal effects would aid in understanding the mechanics of probability distribution being regularized into a low-dimensional manifold.**
>
> **A1**: We guess you mean the top-K regularizer. To further analyze the effects of Top-K regularizer, we conduct the ablation study with different values of regularizer coefficient $\lambda$.  The performance results with different $\lambda$ are shown in Table 4 and the corresponding average  Top-K summations of coreset weights are in Table 5.  In our experiment, there are $50$ candidate samples in each mini-batch data, and the summation of $50$ coreset weights is equal to $1.00$.  Top-K summation of weights increases as $\lambda$ increases, which imposes higher probabilities on the Top-K entries and lower probabilities on the rest candidates. The best performance is achieved when $\lambda=0.1$, which means $\lambda$ balances the trade-off between the loss function and regularizer strength: if $\lambda$ is too large, the algorithm primarily focuses on choosing the important samples instead of updating the model parameter, and vice versa.
>
>
> **Table 4. Ablation study for the regular coefficient $\lambda$**
> | Measure | $\lambda$=0.01 | $\lambda$=0.05| $\lambda$=0.1     | $\lambda$=0.5 | $\lambda$=1   |
> |-------------|--------------------|-------------------|----------------------|------------------|------------------|
> | ACC       | 59.37±0.35     | 60.23±0.43    | **61.60±0.14**  | 59.42±1.45  | 58.89±1.64   |
> | FGT       | 0.095±0.098   | 0.074±0.054 | **0.051±0.015** |0.138±0.075 | 0.128±0.076 |
>
> **Table 5. TopK summation of coreset weights (K=10)**
> | Measure | $\lambda$=0.01 | $\lambda$=0.05 | $\lambda$=0.1 | $\lambda$=0.5 | $\lambda$=1   |
> |-------------|--------------------|--------------------|-------------------|------------------|-----------------|
> | Sum       | 0.41/1.00        | 0.56/1.00        | 0.63/1.00       | 0.73/1.00      | 0.84/1.00    |
>
>
> **Q2: Comparisons with state-of-the-art non-coreset replay methods (e.g., using stability plasticity scores [1], contrastive representation based selection [2]), as it would be meaningful to see the prospect of this direction.**
>
> **[1] Sun et al, Exploring Example Influence in Continual Learning, NeurIPS 2022
> [2] Kim et al, Continual Learning on Noisy Data Streams via Self-Purified Replay, ICCV 2021**
>
> **A2**: We compare two state-of-the-art replay-based methods MetaSP [1] and SPR [2] on Split CIFAR-100, Multiple Datasets, and Tiny-ImageNet. The results are shown in Table 6, 7, and 8 respectively. Our method BCSR outperforms others on balanced, imbalance, and noise settings. In the imbalanced dataset, BCSR demonstrates relatively higher performance.  The reason is that the algorithms in [1] and [2] do not select the most relevant examples in the replay member while our coreset selection method BCSR chooses the most informative samples and saves them into the replay buffer according to the model parameter.
>
> **Table 6. Experiments on Split CIFAR-100**
> | Methods           | Balanced   |             | Imbalanced |             | Label Noise |             |
> |----------------------|---------------|-----------|-----------------|----------|------------------|-----------|
> |                          | AVG ACC |   FGT    | AVG ACC   | FGT     | AVG ACC    | FGT      |
> | MetaSP [1]        | 60.14±0.25 | 0.056±0.23 | 43.74±0.36 | 0.079±0.014 | 57.43±0.54  | 0.086±0.007 |
> | SPR [2]            | 59.56±0.73 | 0.143±0.064 | 44.45±0.55 | 0.086±0.023 | 58.74±0.63  | 0.073±0.010 |
> | BCSR              | **61.60±0.14** | **0.051±0.015** | **47.30±0.57** | **0.022±0.005** | **60.70±0.08**  | **0.059±0.013** |
>
> **Table 7. Experiments on Multiple Datasets**
> | Methods           | Balanced   |             | Imbalanced |             | Label Noise |             |
> |----------------------|---------------|------------|-----------------|----------|------------------|-----------|
> |                          | AVG ACC |   FGT    | AVG ACC   | FGT     | AVG ACC    | FGT      |
> | MetaSP [1]        | 57.14±1.10 | 0.113±0.042 | 41.32±1.50 | 0.103±0.053 | 47.14±1.66  | 0.081±0.027 |
> | SPR [2]            | 56.20±1.91 | 0.124±0.036 | 40.79±1.73 | 0.143±0.051 | 49.77±1.58  | **0.062±0.024** |
> | BCSR              | **59.89±0.95** | **0.096±0.005** | **45.13±0.54** |**0.046±0.008** | **49.97±1.14**  | 0.064±0.031 |
>
> **Table 8. Experiments on Tiny-ImageNet**
> | Methods           | Balanced   |             | Imbalanced |             | Label Noise |             |
> |----------------------|---------------|------------|-----------------|----------|------------------|-----------|
> |                          | AVG ACC |   FGT    | AVG ACC   | FGT     | AVG ACC    | FGT      |
> | MetaSP [1]       | 43.33±0.32 | 0.127±0.002 | 36.75±0.57 | 0.086±0.006 | 37.18±0.76  | 0.068±0.007 |
> | SPR [2]            | 42.79±0.50 | **0.102±0.009** | 36.55±0.74 | 0.070±0.026 | 39.89±0.53  | 0.065±0.021 |
> | BCSR              | **44.13±0.33** | 0.106±0.001 | **38.59±0.11** |**0.047±0.004** | **40.72±0.56**  |**0.055±0.006** |
>
> **Q3: [minor] main illustration Figure 1 could include more technical information (embedding some important eqns for example) or include an additional figure to guide the reader better.**
>
> **A3**: Thank you for your suggestion. We will add a description under the figure in the final version to draw the connection from texts to Equation 1 and pytorch-style pseudocode in Algorithm 1.

---

> > ### Comment · Reviewer_b44u · 2023-08-18
> >
> > Thank you for responding to the concerns.
> > Without some experimental details for SPR and MetaSP, I remain skeptical of the results as the performance gap is minute.

---

> > > ### Author Response · Authors · 2023-08-18
> > > **Experimental details**
> > >
> > > Thanks for your response. Let us provide more details for the experimental settings. MetaSP[1] and SPR[2] show good performance among reply-based methods, but the experimental setups are a bit different due to the coreset selection compared with their original paper.
> > >
> > > 1) The memory buffer of coreset-based is small. Note that the idea of coreset selection is to find the minimal coreset which stands for the most representative examples from the data stream in each task. In SPR, there are two buffers: the delayed buffer D temporarily stocks the incoming data stream, and the purified buffer P maintains the cleansed data. To satisfy the requirement of coreset experiments, we set the size of both buffers as 100 on Split CIFAR-100, 200 on Split Tiny-Imagenet, and 83 on Multiple Dataset. The streaming batch size is set to 10 on Split CIFAR-100 and Multiple Dataset, and 20 on Split Tiny-Imagenet. These buffer sizes and batch sizes follow the settings in coreset-based methods [3,4]. We keep the same buffer and batch size in MetaSP[1], which actually is a big challenge for non-coreset methods.
> > >
> > > 2) Data setting is challenging. The experimental data include balanced, imbalanced, and noise-label. We follow [5] to transform the original dataset to an imbalanced long-tailed CIFAR-100, and set a 20% noise rate for random label shift on three datasets. BCSR outperforms other methods on 3 benchmarks, especially on the imbalanced case (e.g. 6.03% improvement compared to SPR, 8.24% improvement compared to MetaSP on CIFAR-100).
> > >
> > > 3) Details of other hyperparameters. It is worth noting that the training model (called base model in SPR) which is used for evaluation traverses data of the current task only once. The learning rate for model training is set as 0.15 on CIFAR-100, 0.20 on Tiny-ImageNet, and 0.10 on Multiple Datasets. Other experimental parameters for sample selection (including computation of example influence in MetaSP) and buffer updating (e.g., self-centered filtering for current data) are kept as their own settings.
> > >
> > >
> > > [3] Jaehong Yoon, Divyam Madaan, Eunho Yang, and Sung Ju Hwang. Online coreset selection for 489 rehearsal-based continual learning. arXiv preprint arXiv:2106.01085, 2021.
> > >
> > > [4] Borsos, Zalán, Mojmir Mutny, and Andreas Krause. "Coresets via bilevel optimization for continual learning and streaming." Advances in neural information processing systems 33 (2020): 14879-14890.
> > >
> > > [5]Yin Cui, Menglin Jia, Tsung-Yi Lin, Yang Song, and Serge Belongie. Class-balanced loss based on effective number of samples. In Proceedings of the IEEE/CVF conference on computer vision and pattern recognition, pages 9268–9277, 2019.

---

### Official Review · Reviewer_vGFs · 2023-07-02

**Soundness:** 3 good
**Presentation:** 3 good
**Contribution:** 2 fair
**Rating:** 5
**Confidence:** 4

**Summary:**

The authors present a new approach to coreset selection in rehearsal-based continual learning. The authors claim that traditional methods optimise over discrete decision variables, resulting in computationally expensive processes. To address this, they propose a new bilevel formulation where the inner problem finds a model that minimizes expected training error, and the outer problem learns a probability distribution with approximately K nonzero entries through adding a regularizer and ensuring convergence to the ϵ-stationary point with O(1/ϵ^4) complexity. The authors also perform extensive experiments demonstrate superior performance compared to existing methods in various continual learning settings.

**Strengths:**

1. The availability of provided code facilitates reproducibility, making it a straightforward process.
2. The authors present experimental results that demonstrate a pretty good performance in comparison to the selected baselines.
3. The inclusion of the added regularizer seems promising, but it requires a more detailed and elaborate explanation to justify its contribution adequately.

**Weaknesses:**

**Weaknesses**
My main concern is that this work appears to be primarily an incremental extension of prior work [2] by adding a regularizer. While this is not necessarily negative, I do not see any significant new improvements from an algorithmic perspective. Specifically, in lines 46-52, the authors assert that the key challenges of bilevel coreset selection are (1) the expensive nature of optimization over cardinality constraints and (2) the nested nature of bilevel optimization. However, these weaknesses pertain to [1] rather than the coreset selection problem itself.
In order to address these challenges, [2] was proposed. The contributions that the authors claim in this paper seem to stem from [2] rather than the proposed method itself.

**Suggestion:**
Certainly, enhancing the motivation of the paper and clarifying the advantages of the proposed method compared to [7] is crucial for improving the paper. The author should allocate more space to address these aspects.

[1] Borsos, Z., Mutny, M., & Krause, A. (2020). Coresets via bilevel optimization for continual learning and streaming. Advances in neural information processing systems, 33, 14879-14890.

[2] Zhou, X., Pi, R., Zhang, W., Lin, Y., Chen, Z., & Zhang, T. (2022, June). Probabilistic bilevel coreset selection. In International Conference on Machine Learning (pp. 27287-27302). PMLR.

**Questions:**

Could you please clarify the differences and advantages between your method and [7]? In lines 135-139, the author simply states that "so this formulation [7] oversimplifies the coreset selection problem." However, this statement does not highlight any specific disadvantages of [7].

I will consider improving my score if my question and concern could be well-addressed.

---

> ### Author Rebuttal · Authors · 2023-08-08
>
> Thanks for your reviews. We have addressed your concerns below.
>
> **Q1: Could you please clarify the differences and advantages between your method and [2]? In lines 135-139, the author simply states that "so this formulation [2] oversimplifies the coreset selection problem." However, this statement does not highlight any specific disadvantages of [2].**
>
> **[1] Borsos, Z., Mutny, M., & Krause, A. (2020). Coresets via bilevel optimization for continual learning and streaming. Advances in neural information processing systems, 33, 14879-14890.**
>
> **[2] Zhou, X., Pi, R., Zhang, W., Lin, Y., Chen, Z., & Zhang, T. (2022, June). Probabilistic bilevel coreset selection. In International Conference on Machine Learning (pp. 27287-27302). PMLR.**
>
>
> **A1**: Thank you for this insightful question. We will improve our presentation, especially the comparison with [2]. The challenges of bilevel continual learning in [1] are:  1) performing optimization over discrete decision variables with greedy search; 2) the nested structure of bilevel problems. Both [2] and our method BCSR relax the discrete bilevel optimization problem into a continuous one. Our approach improves over [2] due to the following differences.
>
> 1) Zhou et al. [2] relaxes the bilevel formulation to minimize the loss function over the Bernoulli distribution $s$, i.e., $\min_{s\in \mathcal{C}} \Phi(s)$,  and develop a policy gradient solver to optimize the Bernoulli variable. Their gradient $\nabla_{s}\Phi(s) = E_{p(m|s)}\Big[L(\theta^*(m))\nabla_{s}\ln p(m|s)\Big]$ discards the implicit gradient of $L(\theta^*(m))$ in terms of $s$. However, $\theta^*(m)$ actually depends on the mask $m$, and $m$ depends on the Bernoulli variable $s$. Therefore there is an implicit gradient because changes in $s$ will cause changes in $\theta^*(m)$: this fact is ignored by Zhou et al. [2] and hence it is a disadvantage. In contrast, our bilevel optimization computes the hypergradients for the coreset weights $w$ ($0\leq w \leq 1$ and $\|w\|_1 =1$ ), which considers the implicit dependence between $\theta^*(w)$ and $w$. The experiment results also prove that our method is much better than Zhou et al. [2] for coreset selection.
>
> 2) Zhou et al. [2] assumes that the inner loop converges to $\theta^*(m)$ exactly, which oversimplifies the analysis and may not hold in practice. In contrast, we carefully analyze the gap between estimated $\hat{\theta}(m)$ by our algorithm and the minimizer of the inner problem $\theta^*(m)$. Please note that we use $m$ (i.e., sample mask) in the rebuttal to follow the notation in Zhou et al. [2] but our paper uses notation $w$ to denote coreset weights (which is equivalent to sample mask).
>
> 3) Our new bilevel formulation introduces a novel smoothed Top-K regularizer, which is important as shown in our ablation studies (Table 6 and Section G in the original manuscript). In contrast, [2] does not use such a regularizer. Indeed, we show empirically that our algorithm BCSR is  better than Zhou et al. [2] in a wide spectrum of benchmark datasets, as illustrated in Table 1~4 in the main text.

---

> ### Author Response · Authors · 2023-08-15
> **Looking forward to post-rebuttal feedback!**
>
> Dear Reviewer vGFs,
>
> Thank you for reviewing our paper. We have carefully addressed your concerns regarding the clarification of the comparison with the reference [7]. Please let us know if our response addressed your concerns about the presentation. If our response resolves your concerns, we kindly ask you to consider raising the rating of our work. Thank you very much! We are happy to discuss any additional questions you may have.
>
> [7] Zhou, X., Pi, R., Zhang, W., Lin, Y., Chen, Z., & Zhang, T. (2022, June). Probabilistic bilevel coreset selection. In International Conference on Machine Learning (pp. 27287-27302). PMLR.
>
>
> Best,
> Authors

---

> > ### Comment · Reviewer_vGFs · 2023-08-18
> >
> > Thanks for your feedback. I will raise my score. I think most of my concerns are addressed.

---

### Official Review · Reviewer_AXy9 · 2023-07-03

**Soundness:** 3 good
**Presentation:** 3 good
**Contribution:** 3 good
**Rating:** 7
**Confidence:** 3

**Summary:**

This work addresses the coreset selection problem in rehearsal-based continual learning, focusing specifically on the application of bilevel optimization. The authors identify limitations in existing bilevel optimization-based coreset selection methods for continual learning, including high computational costs resulting from greedy search and the loss of bilevel optimization nature due to single-level equivalence. To overcome these drawbacks, the authors propose a new formulation that incorporates the probability simplex and a smoothed top-K regularizer. The latter enforces the K most important elements have larger weights. The authors develop a new stochastic bilevel method for this continuous and smooth loss. The effectiveness of the proposed method is demonstrated through comprehensive experiments, which also include various ablation studies. The authors establish the properties of the new loss function and provide guarantees on convergence.

**Strengths:**

1. Both continual learning and bilevel optimization are timely topics.The exploration of the benefits of bilevel optimization in continual learning is an under-explored area, making the studied topic in this work both interesting and important.
2. The paper is well written and easy to follow. The inclusion of Figure 1, illustrating the process of training bilevel algorithms in continual learning, along with PyTorch-style pseudocode, greatly aids in understanding the entire process.
3. The introduction of the new loss function and bilevel algorithms is well-motivated. The replacement of the nuclear norm with a smoothed top-K regularizer, as proposed in the paper, is a good idea that leads to faster and improved bilevel optimization algorithms. The empirical evidence provided in Figure 3 serves as a strong justification for the efficacy of the new loss function and algorithms.
4. The experiments conducted in the paper are comprehensive. The evaluation encompasses various datasets, including multiple datasets, split Cifar100, and large-scale datasets like Tiny-ImageNet and Food-101. Additionally, various ablation studies are also included.
5. The paper establishes guarantees on convergence rate and explores the properties of the new loss function.

**Weaknesses:**

The proposed algorithm needs to compute the Hessian-vector products, which may be computation expensive in large-scale cases. This computational expense may pose challenges and limit the applicability of the algorithm. Is it possible to improve the efficiency of the proposed method by utilizing Hessian-free bilevel algorithms based on recent advancements (e.g., Liu et al. [1], Sow et al. [2])？

[1] Liu, Bo, et al. "Bome! bilevel optimization made easy: A simple first-order approach." Advances in Neural Information Processing Systems 35 (2022): 17248-17262.

[2] Sow, Daouda, Kaiyi Ji, and Yingbin Liang. "On the convergence theory for hessian-free bilevel algorithms." Advances in Neural Information Processing Systems 35 (2022): 4136-4149.



**Questions:**

1. How to select the parameters such as N, Q and $\delta$ in Algorithm 3 for continual learning? It would be beneficial to provide suggestions or guidelines for choosing these parameters and to discuss the sensitivity of the algorithm's performance to such parameters.

2. What are the main challenges of the bilevel optimization analysis in the continual learning setting?

**Limitations:**

yes

---

> ### Author Rebuttal · Authors · 2023-08-08
>
> Dear Reviewer AXy9,
>
>  Thanks for your reviews. We have addressed your concerns below.
>
> **Q1: Is it possible to improve the efficiency of the proposed method by utilizing Hessian-free bilevel algorithms based on recent advancements?**
>
> **A1**: Yes, two types of Hessian-free methods may be considered. The first approach is to approximate the Hessian- and Jacobian-vector products via finite-difference estimation, i.e., using $[\nabla f(x+\alpha v)-\nabla f(x-\alpha v)]/(2\alpha)$ to approximate $\nabla^2 f(x)v$. Then, the entire process will not contain second-order information. The second approach is to apply the recent value-function-based problem reformulation and constrained optimization-based approaches [1]. We will investigate these ideas in future studies.
>
> [1] B. Liu et al. “Bome! bilevel optimization made easy: A simple first-order approach.” NeurIPS 2022.
>
> **Q2: How to select the parameters such as N, Q, and $\delta$ in Algorithm 3 for continual learning? It would be beneficial to provide suggestions or guidelines for choosing these parameters and to discuss the sensitivity of the algorithm's performance to such parameters.**
>
> **A2**: We conduct ablation studies to explore the sensitivity of hyperparameters ($N$ and $Q$) on  Split CIFAR-100. The results are presented in Tables 2 ($N$) and 3 ($Q$). The model performance remains relatively stable when increasing inner loops ($N$) while fixing $Q$.  But too large $N$ ($N\geq 15$) leads to performance degradation due to overfitting.  The $Q$ loops show similar properties that a few $Q$ loops (e.g., $Q=3$) are enough to approximate the Hessian inverse product. Too small $Q$ or too large $Q$ will hurt the performance due to possible underfitting (e.g., $Q=1$) or overfitting (e.g., $Q=20$). $\delta$ is a factor for Gaussian noise. It is usually small enough for Gaussian smoothness, so it is fixed as $1e-3$ in the experimental setting.
>
> **Table 2.  Ablation study for the inner loops (N) with fixed loops $Q=3$.**
> | Measure | N=1             | N=5              | N=10           | N=15            | N=20            |
> |-------------|------------------|------------------|-----------------|------------------|------------------|
> | ACC       | 61.60±0.14  | 61.75±0.11   | 61.64±0.15  | 60.77±0.32   | 59.20±0.41  |
> | FGT       | 0.051±0.015| 0.047±0.013| 0.063±0.017 | 0.074±0.021 | 0.079±0.035 |
>
> **Table 3. Ablation study for the loops Q with fixed inner loops $N=1$.**
> | Measure |  Q=1                 | Q=3            | Q=5               | Q=10            | Q=20            |
> |-------------|----------------------|-----------------|-------------------|------------------|------------------|
> | ACC       | 52.14±1.53       | 61.60±0.14  | 61.57±0.15   | 58.42±0.53   |  57.80±1.31  |
> | FGT       |  0.123±0.038    |0.051±0.015 | 0.064±0.012  | 0.173±0.045 | 0.162±0.041 |
>
> **Q3: What are the main challenges of bilevel optimization analysis in the continual learning setting?**
>
> **A3**: Existing works on bilevel optimization mainly focus on the unconstrained case, and the analysis often relies on the continuity between two adjacent iterates. As a comparison, our analysis in CL setting needs to deal with the constraint over the simplex, and it is non-trivial to handle the possible discontinuity caused by the projection.  Another challenge lies in showing the smoothness of the problem as well as a hypergradient estimation error under the stochastic Top-K regularizer.

---

> > ### Comment · Reviewer_AXy9 · 2023-08-17
> >
> > Thanks for the response.

---

### Official Review · Reviewer_FPqc · 2023-07-03

**Soundness:** 3 good
**Presentation:** 3 good
**Contribution:** 3 good
**Rating:** 7
**Confidence:** 3

**Summary:**

The paper presents a better approach for grasping the coreset-based bi-level optimization procedure used in the field of continual learning. The proposed method relies on the use of the proxy model to obtain a coreset, which in turn will affect the training of the original model.  With that being said, the idea relies on relaxing the hard cardinality constraints usually used in the field of coreset construction for continual learning tasks, to a softer version that allows the use of first-order methods.

**Strengths:**

In what follows, a list of the strengths of the presented paper is given:

 * The paper draws back from the hard cardinality constraint and follows a relaxed version of it by resolving to the use of probability simplex and the top-K smoothed regularization to mimic the effect of the lost cardinality constraint.
 * The method is easy to follow, as well as it relies on the use of a proxy model.
 * While the proposed model is among the top 5 slowest from the list of competitors (referring to Table 8 in the appendix), it is by far the best method in terms of accuracy and forgetting almost across the board.
 * The new formulation of the problem allows for dropping the use of combinatorial optimization techniques while exposing the field to first-order optimizers.
 * Finally, the paper is well-written and easy to follow.

**Weaknesses:**

One problem that I have in mind is the use of the proxy model, which might be a limitation in the presence of large models, i.e., models containing $> 100M$ parameters.

**Questions:**

Is it possible to circumvent the need for a proxy model in the case of working with large models? How worse would the results be if one would use some quantized or pruned version of the original model $M_{tr}$?

**Limitations:**

The authors listed the limitations associated with their theoretical analysis in the form of assumptions usually used in the field of coresets for continual learning.

---

> ### Author Rebuttal · Authors · 2023-08-08
>
> Dear Reviewer FPqc:
>
> Thanks for your reviews. We have addressed your concerns below.
>
> **Q1: The proxy model $M_{cs}$ might be a limitation in the presence of large models, i.e., models containing >100M  parameters. Is it possible to circumvent the need for a proxy model in the case of working with large models? How worse would the results be if one would use some quantized or pruned version of the original model $M_{tr}$?**
>
> **A1**: It is feasible to use only one model for the training and coreset selection phases. After each training phase, the model parameters can be saved in the disk temporarily and then the model continues to conduct coreset selection.  In the next model training phase, the saved model parameters are loaded into the current model again. This method can handle large models but with an extra overload of loading and checkpointing.
>
> In addition, we apply the quantization technique [1] to the models $M_{cs}$ and $M_{tr}$ with lower bits. Specifically, we keep the full precision of the model $M_{tr}$ during the learning of each task, but the model  $M_{tr}$ is quantized upon finishing one task. In particular, the gradients are calculated on quantized model parameters (INT16, INT8 and INT4) to perform gradient-based updates. After several steps of updates upon finishing the training of one task, the full precision model is quantized and saved. Then the quantized model is loaded to $M_{cs}$ for coreset selection. The above process repeats until finishing all the tasks. Finally, the quantized model $M_{tr}$ is used for performance evaluation. The results (ACC, FGT, quantized model size) are shown in Table 1. Specifically, models are quantized to INT16, INT8 and INT4 respectively, but with a small performance degradation (e.g., reduced by up to $4.1$% for INT4). However, the model size shrinks from $3.76$MB to $1.67$MB (INT16), $0.93$MB (INT8) and $0.61$MB (INT4), respectively, which reduces the memory footprint significantly (e.g., up to $83.8$% decrease for INT4). The results demonstrate that the quantized model achieves comparable performance to the original full precision model for the coreset selection but with a much smaller model size. That provides some useful insights into a quantized version of a large model applied in continual learning.
>
> **Table 1. The performance of the quantized model of ResNet18 on Split CIFAR-100 (FP32 means the original model with full precision).**
> | Param Type | ACC             | FGT                | Model Size (MB)|
> |------------------|------------------|--------------------|-----------------------|
> | FP32            | 61.60±0.14  | 0.051±0.015    |  3.76                  |
> | INT16           | 60.03±0.16  | 0.064±0.017    |  1.67                  |
> | INT8             | 59.12±0.21  | 0.081±0.024    |  0.93                  |
> | INT4             | 59.06±0.15  | 0.085±0.027    |   0.61                 |
>
> [1] Polino A, Pascanu R, Alistarh D. Model compression via distillation and quantization[J]. arXiv preprint arXiv:1802.05668, 2018.

---

> > ### Comment · Reviewer_FPqc · 2023-08-16
> >
> > Thanks for clarifying my concern.

---

### Author Rebuttal · Authors · 2023-08-08

**General Response:**

We would like to thank all reviewers for their constructive comments. We have answered the corresponding questions from reviewers and provided new experiment results requested by reviewers. The main summary of our responses includes:

1. Per Reviewer FPqc suggestion:
	- We have added the quantized experiments for models $M_{tr}$ and $M_{cs}$, and compared the performance (ACC and FGT) between quantized models and original models in continual learning. The descriptions of experimental details are included in our response. We show that our quantized models do not lose too much performance but require much smaller memory.

2. Per Reviewer AXy9 suggestion:
      - We have cited and discussed Hessian-free methods, which can be used to improve the efficiency of our method.
	- We have conducted ablation experiments for hyperparameters $N$, $Q$ for sensitivity analysis, and stated the details for $\delta$ setting.
	- We have explained the main challenges of bilevel optimization analysis in continual learning: 1) bilevel optimization on the constrained case; 2) smoothness of the problem.

3. Per Reviewer vGFs suggestion:
	- We have clarified the main differences and advantages between our method and Zhou et al. [ref1].

[ref1] Zhou, X., Pi, R., Zhang, W., Lin, Y., Chen, Z., & Zhang, T. (2022, June). Probabilistic bilevel coreset selection. In International Conference on Machine Learning (pp. 27287-27302). PMLR.

4. Per Reviewer b44u suggestion:
	- We have further analyzed the effects of Top-K regularizer and conducted ablation experiments for regular coefficient $\lambda$.
	- We have added experiments for comparisons with state-of-the-art non-coreset replay methods [ref2] and [ref3]: the results are included in new tables in the rebuttal.
	- We will add a description under the figure in the final version to draw the connection from texts to equation 1 and pytorch-style pseudocode in Algorithm 1.

[ref2] Sun et al, Exploring Example Influence in Continual Learning, NeurIPS 2022

[ref3] Kim et al, Continual Learning on Noisy Data Streams via Self-Purified Replay, ICCV 2021

---

### Decision · Program_Chairs · 2023-09-21

**Decision:**

Accept (poster)

**Comment:**

The paper proposes a new algorithm for the problem of coreset selection for continual learning. The authors propose a new formulation of the problem, demonstrate a new bi-level optimization procedure for this formulation, and conduct multiple experiments to support their claims.
The reviewers commended the novelty and simplicity of the approach, the use of probability simplex and top-k regularization and the thoroughness of the experiments. During the discussion period, most of the concerns have been addressed. There is consensus among the reviewers that the paper should be accepted.
After reading the paper, the reviews and the discussions, I feel that this is a solid work and the community will benefit from understanding this result. Thus, I recommend acceptance.